# Osteocytic oxygen sensing controls bone mass through epigenetic regulation of sclerostin

Steve Stegen[1], Ingrid Stockmans[1], Karen Moermans[1], Bernard Thienpont [2], Patrick H. Maxwell[3], Peter Carmeliet[4,5] & Geert Carmeliet [1]

Preservation of bone mass is crucial for healthy ageing and largely depends on adequate responses of matrix-embedded osteocytes. These cells control bone formation and resorption concurrently by secreting the WNT/β-catenin antagonist sclerostin (SOST). Osteocytes reside within a low oxygen microenvironment, but whether and how oxygen sensing regulates their function remains elusive. Here, we show that conditional deletion of the oxygen sensor prolyl hydroxylase (PHD) 2 in osteocytes results in a high bone mass phenotype, which is caused by increased bone formation and decreased resorption. Mechanistically, enhanced HIF-1α signalling increases Sirtuin 1-dependent deacetylation of the *Sost* promoter, resulting in decreased sclerostin expression and enhanced WNT/β-catenin signalling. Additionally, genetic ablation of PHD2 in osteocytes blunts osteoporotic bone loss induced by oestrogen deficiency or mechanical unloading. Thus, oxygen sensing by PHD2 in osteocytes negatively regulates bone mass through epigenetic regulation of sclerostin and targeting PHD2 elicits an osteo-anabolic response in osteoporotic models.

[1] Laboratory of Clinical and Experimental Endocrinology, Department of Chronic Diseases, Metabolism and Ageing, KU Leuven, 3000 Leuven, Belgium. [2] Laboratory for Functional Epigenetics, Department of Human Genetics, KU Leuven, 3000 Leuven, Belgium. [3] Cambridge Institute for Medical Research, University of Cambridge, Cambridge CB2 0XU, UK. [4] Laboratory of Angiogenesis and Vascular Metabolism, Center for Cancer Biology, Department of Oncology, KU Leuven, 3000 Leuven, Belgium. [5] Laboratory of Angiogenesis and Vascular Metabolism, Center for Cancer Biology, VIB, 3000 Leuven, Belgium. Correspondence and requests for materials should be addressed to G.C. (email: geert.carmeliet@kuleuven.be)

Bone needs to be continuously remodelled throughout life to maintain optimal quality and strength. This process requires a strict balance between the activities of bone-forming osteoblasts and bone-resorbing osteoclasts to avoid bone loss[1]. Recent findings show that terminally differentiated matrix-embedded osteocytes are crucial in the regulation of bone remodelling, because they can control both osteoblast and osteoclast differentiation and function[2–6]. An important factor in the communication of osteocytes with osteoblasts and osteoclasts is sclerostin, a secreted WNT/β-catenin antagonist encoded by the *Sost* gene[7,8]. Genetic evidence shows that increased sclerostin levels result in decreased bone formation and increased bone resorption[9–15], suggesting that factors regulating sclerostin expression are likely as critical for bone homoeostasis.

The control of bone remodelling by osteocytes is influenced by endocrine and paracrine factors such as parathyroid hormone and WNTs[4–6], but possibly also by the local environment. Indeed, osteocytes embedded in the cortical bone matrix are exposed to low oxygen tensions[16], but whether osteocytic oxygen sensing is involved in the control of bone homoeostasis is still unknown. In general, cells can respond to hypoxia by an elegant pathway consisting of prolyl hydroxylases (PHDs) that detect fluctuations in oxygen tension and regulate the abundance of the hypoxia-inducible transcription factor HIF-α, which is considered as the effector of the hypoxia response. When oxygen levels fall below a critical threshold, PHD activity is diminished, resulting in HIF stabilization, which induces transcription of genes involved in diverse physiological processes ranging from angiogenesis to metabolism to matrix synthesis[17–19]. In fact, in vivo oxygen tensions in osteocyte lacunae are reported to be well below 10%[16], a critical threshold for PHD enzyme activity in vitro[20,21], suggesting that stringent regulation of HIF signalling may be necessary for proper osteocyte functioning.

We and others have already shown that enhanced HIF signalling in osteoprogenitors and osteoblasts increases bone mass during development, homoeostasis, regeneration and pathology[22–28], and that the opposite phenotype is found when *Hif-1α*

is deleted in osteolineage cells[22,29,30]. The high bone mass phenotype is explained by increased angiogenesis or a metabolic shift to glycolysis, although detailed understanding of the bone anabolic response is still lacking. In addition, some recent studies show that PHD activity in osteoprogenitors is involved in the HIF-induced increase in bone mass, although the contribution of the different PHDs and the underlying mechanism are not fully elucidated yet: constitutive combined deletion of *Phd2* and *Phd3* resulted in HIF-2α stabilization and decreased osteoclastogenesis[26], whereas inducible deletion of *Phd2* was sufficient to increase bone mass, although the mechanism was not further investigated[27,28]. Given the importance of osteocyte function for bone metabolism and the fact that they reside in a low oxygen environment, we hypothesized that oxygen sensing in osteocytes controls bone homoeostasis and may interfere with pathological bone loss.

We therefore investigated whether and how PHD oxygen sensors control bone mass by generating mice with osteocyte-specific genetic ablation of PHD2, the most abundantly expressed isoform. We discovered that osteocytic deletion of PHD2 controls bone formation and resorption through a Sirtuin 1 (SIRT1)-dependent decrease in sclerostin expression resulting in increased WNT/β-catenin signalling. Moreover, mice lacking PHD2 in osteocytes are protected from disuse- and oestrogen deficiency-induced bone loss, suggesting that therapeutic targeting of PHD2 can potentially be used for the treatment of osteoporosis.

## Results

**Osteocyte-specific PHD2 deletion increases bone mass.** To explore whether the oxygen sensor PHD2 is critical for osteocyte function, we crossed *Phd2^{fl/fl}* mice with *Dentin Matrix Protein 1* (*Dmp1*)-*Cre* transgenic mice (*Phd2^{ot−}*) resulting in very efficient and specific deletion (Fig. 1a and Supplementary Fig. 1a-c). Of note, *Phd2* was highly expressed in osteocytes compared to other osteogenic cells (Supplementary Fig. 1a, b), underscoring the importance of the oxygen sensing machinery in osteocytes. Yet, oxygen levels seem still adequate, as activation of hypoxia

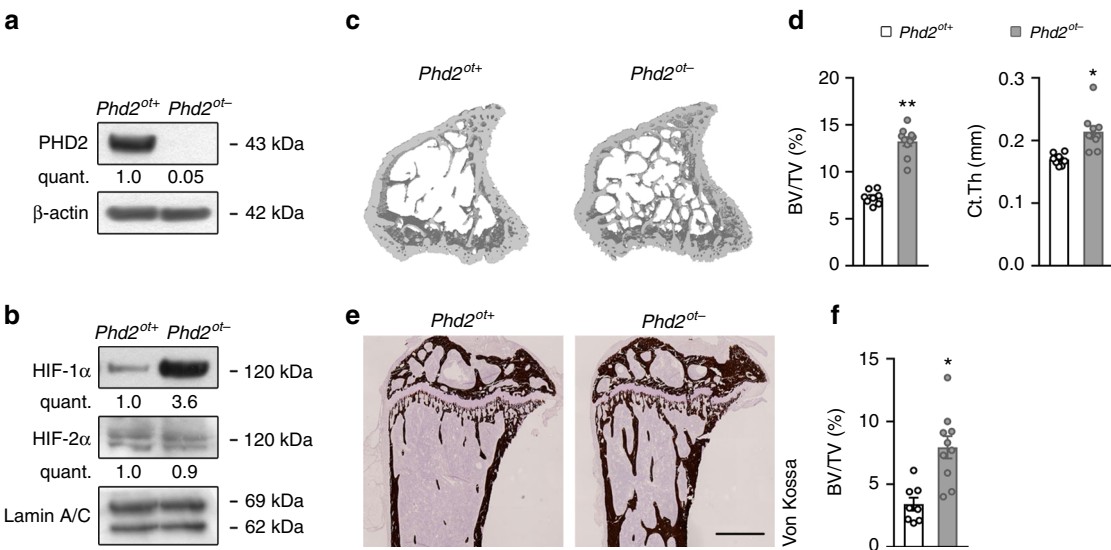

**Fig. 1** Deletion of PHD2 in osteocytes increases bone mass. **a** PHD2 and β-actin immunoblot on whole-cell extracts isolated from osteocyte-enriched bone fractions of 8-week-old mice. Results are representative of three experiments. **b** HIF-1α, HIF-2α and Lamin A/C immunoblot on nuclear cell extracts isolated from osteocyte-enriched bone fractions. Results are representative of three experiments. **c**, **d** 3D microCT models (**c**) of the tibial metaphysis and quantification (**d**) of trabecular bone volume (BV/TV) and cortical thickness (Ct.Th) in 8-week-old mice ($n = 8$ *Phd2^{ot+}*–10 *Phd2^{ot−}*). **e**, **f** Von Kossa staining (**e**) of tibiae with quantification (**f**) of trabecular bone volume (BV/TV) ($n = 8$ *Phd2^{ot+}*–10 *Phd2^{ot−}*). Scale bar is 500 μm. Data are means ± SEM. *$p < 0.05$ vs. *Phd2^{ot+}*, **$p < 0.01$ vs. *Phd2^{ot+}* (Student's *t*-test)

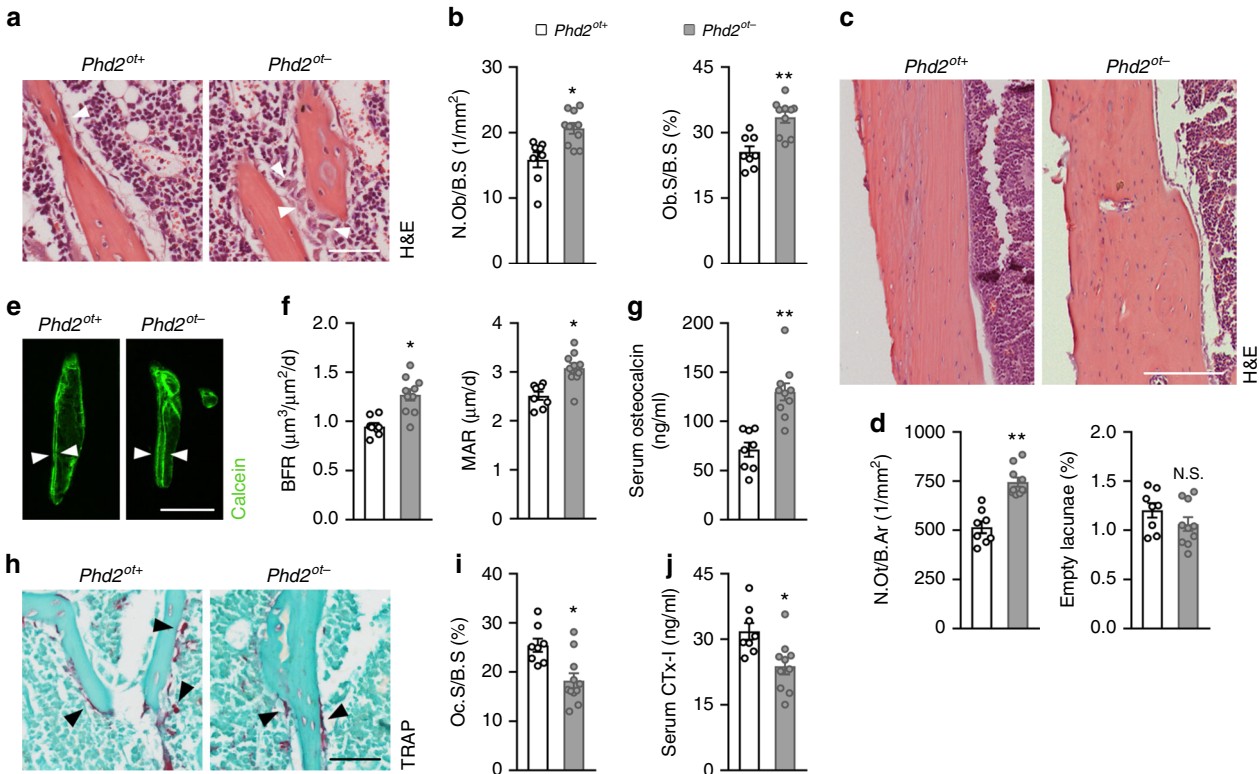

**Fig. 2** Bone formation exceeds bone resorption in $Phd2^{ot-}$ mice. **a**, **b** H&E staining (**a**) of the tibial metaphysis with quantification (**b**) of osteoblast number per bone surface (N.Ob/B.S) and osteoblast surface per bone surface (Ob.S/B.S) in 8-week-old mice ($n = 8$ $Phd2^{ot+}$–10 $Phd2^{ot-}$). White arrowheads in **a** indicate osteoblasts. **c**, **d** H&E staining (**c**) of the cortical diaphysis of tibiae with quantification (**d**) of osteocyte number per bone area (N.Ot/B.Ar) and percentage of empty osteocyte lacunae ($n = 8$ $Phd2^{ot+}$–10 $Phd2^{ot-}$). **e**, **f** Calcein labelling of mineralizing surfaces on trabeculae (**e**) with quantification (**f**) of the bone formation rate (BFR) and mineral apposition rate (MAR) ($n = 8$ $Phd2^{ot+}$–10 $Phd2^{ot-}$). White arrowheads in **e** indicate calcein incorporation. **g** Serum osteocalcin levels ($n = 8$ $Phd2^{ot+}$–10 $Phd2^{ot-}$). **h**, **i** TRAP staining (**h**) of the tibial metaphysis with quantification (**i**) of osteoclast surface per bone surface (Oc.S/B.S) ($n = 8$ $Phd2^{ot+}$–10 $Phd2^{ot-}$). Black arrowheads in **h** indicate osteoclasts. **j** Serum CTx-I levels ($n = 8$ $Phd2^{ot+}$ –10 $Phd2^{ot-}$). Data are means ± SEM. *$p < 0.05$ vs. $Phd2^{ot+}$, **$p < 0.01$ vs. $Phd2^{ot+}$, N.S. is not significant (Student's t-test). Scale bars in **a**, **e** and **h** are 50 μm, scale bar in **c** is 100 μm

signalling was largely absent in $Phd2^{ot+}$ mice, evidenced by the low number of HIF-1α-positive nuclei (Supplementary Fig. 1d). Deficiency of PHD2 resulted in increased nuclear accumulation of HIF-1α, but not HIF-2α (Fig. 1b and Supplementary Fig. 1d). $Phd2^{ot-}$ mice were viable and undistinguishable from control littermates at birth, and showed normal growth, as body weight and tibia length were comparable to control mice at 8 weeks of age (Supplementary Fig. 1e, f). Of note, we could not observe any sign of erythrocytosis (Supplementary Fig. 1g–j), which has been previously reported when HIF signalling was induced in osteoblasts[31].

Deletion of PHD2 in osteocytes increased bone mass with 40% in 3-week-old male and female mice (Supplementary Fig. 2a, b) and this increase in trabecular and cortical bone volume became more pronounced in adult mice and was preserved with ageing, as analysed by ex vivo micro-computed tomography (microCT) and histomorphometry on Von Kossa-stained sections (Fig. 1c–f and Supplementary Fig. 2a, b). In correspondence with the enhanced bone mass, bone strength was increased in $Phd2^{ot-}$ mice, as evidenced by three-point bending tests (Supplementary Fig. 2c–f). Thus, deletion of PHD2 in osteocytes causes an increase in bone mass, indicating that bone formation exceeded resorption.

**Bone formation exceeds bone resorption in $Phd2^{ot-}$ mice**. To investigate the cellular mechanisms responsible for the high bone mass phenotype of $Phd2^{ot-}$ mice, we first analysed bone

formation parameters. Histomorphometry of H&E-stained sections revealed an increase in the number and surface of cuboidal osteoblasts lining the trabeculae (Fig. 2a, b) and in the density of viable osteocytes in the cortical bone of 8-week-old male $Phd2^{ot-}$ mice (Fig. 2c, d). In addition, we observed enhanced osteoblast activity in $Phd2^{ot-}$ mice, evidenced by an increase in the thickness of unmineralized matrix (osteoid) and in its mineralization, as analysed by Goldner staining (Supplementary Fig. 3a, b) and in vivo calcein labelling (Fig. 2e, f), respectively. Consistent with these observations, we found an increase in serum osteocalcin levels (Fig. 2g) and femoral transcript levels of osteoblast markers including runt-related transcription factor 2 ($Runx2$), collagen type I ($Col1$) and osteocalcin ($Ocn$) in $Phd2^{ot-}$ mice (Supplementary Fig. 3c).

In addition, we noticed that silencing PHD2 in osteocytes also decreased bone resorption. Histomorphometric analysis of TRAP-stained sections revealed a reduction in osteoclast surface in mutant mice (Fig. 2h, i), and the decrease in serum collagen type I cross-linked C-telopeptide (CTx-I) levels underscores the reduction in osteoclast activity (Fig. 2j). Interestingly, hematopoietic bone marrow cells of $Phd2^{ot-}$ mice were able to differentiate properly to osteoclasts in vitro (Supplementary Fig. 3d, e), suggesting that the decrease in bone resorption is due to a cell non-autonomous effect. Taken together, silencing PHD2 in osteocytes causes a high bone mass phenotype, resulting from the combination of increased bone formation and decreased bone resorption.

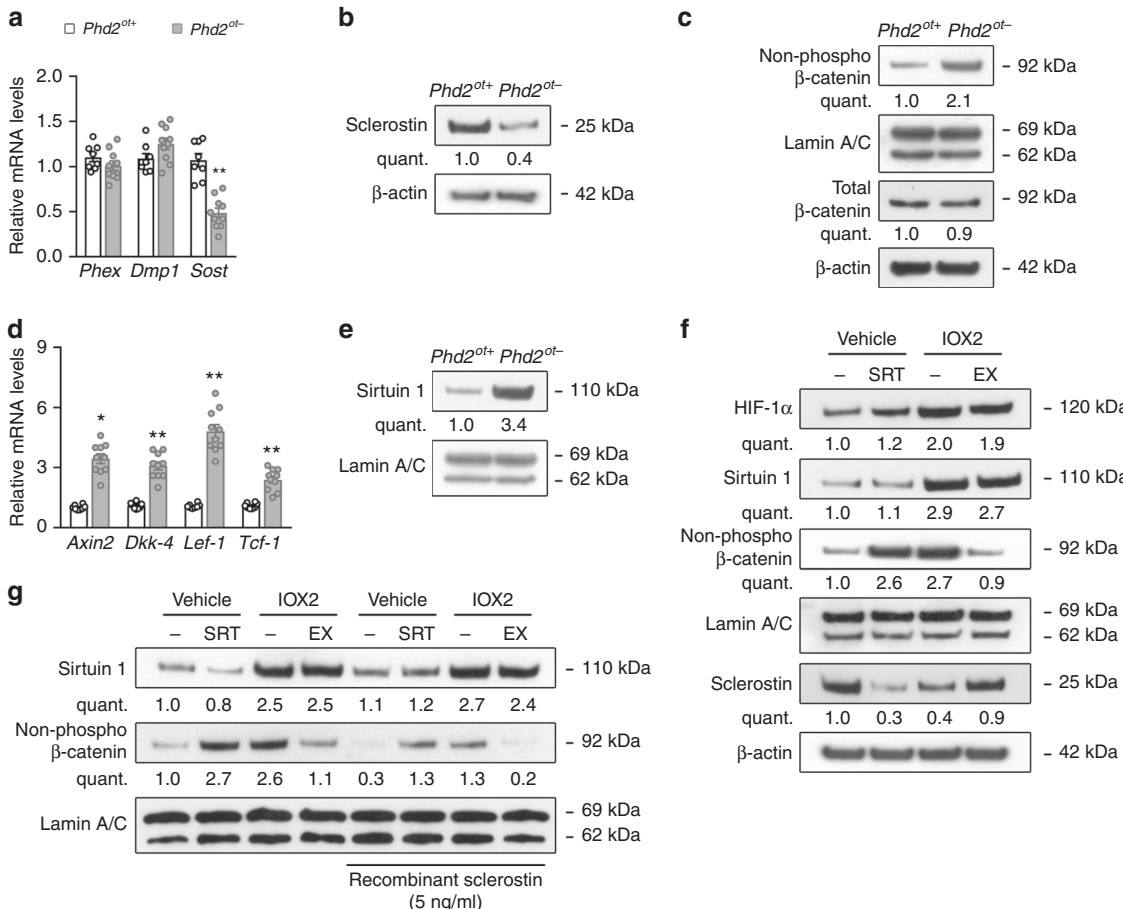

**Fig. 3** PHD2 deletion reduces sclerostin expression through SIRT1. **a** *Phex*, *Dmp1* and *Sost* mRNA levels in osteocyte-enriched bone fractions of 8-week-old mice (*n* = 8 *Phd2*$^{ot+}$–10 *Phd2*$^{ot−}$). **b** Sclerostin and β-actin immunoblot on whole-cell extracts isolated from osteocyte-enriched bone fractions. **c** Non-phosphorylated (non-phospho) β-catenin and Lamin A/C immunoblot (nuclear cell extracts), and total β-catenin and β-actin immunoblot (whole-cell extracts). Protein extracts are isolated from osteocyte-enriched bone fractions. **d** *Axin2*, *Dkk-4*, *Lef-1* and *Tcf-1* mRNA levels in osteocyte-enriched bone fractions (*n* = 8 *Phd2*$^{ot+}$– 10 *Phd2*$^{ot−}$). **e** Sirtuin 1 and Lamin A/C immunoblot on nuclear cell extracts isolated from osteocyte-enriched bone fractions. **f** Sirtuin 1, non-phospho β-catenin, Lamin A/C, sclerostin and β-actin immunoblot isolated from vehicle (IDG$^{VEH}$) or IOX2-treated IDG-SW3 (IDG$^{IOX2}$) cells after 14 days of osteogenic differentiation. IDG$^{VEH}$ cells were treated with vehicle (−) or SRT1720 (SRT); IDG$^{IOX2}$ cells were treated with vehicle (−) or EX527 (EX). **g** Sirtuin 1, non-phospho β-catenin and Lamin A/C immunoblot on nuclear cell extracts isolated from IDG$^{VEH}$ or IDG$^{IOX2}$ cells after 14 days of osteogenic differentiation in the presence of recombinant sclerostin. Cells were treated as in **f**. Data are means ± SEM. Immunoblot images are representative of three experiments. *$p < 0.05$ vs. *Phd2*$^{ot+}$, **$p < 0.01$ vs. *Phd2*$^{ot+}$ (Student's *t*-test)

**PHD2 deletion decreases sclerostin expression through SIRT1.**
The opposite changes in bone formation and bone resorption elicited by PHD2 deletion in osteocytes suggested the involvement of sclerostin, a WNT/β-catenin inhibitor. Indeed, sclerostin mRNA and protein levels were decreased in osteocytes of *Phd2*$^{ot−}$ long bones (Fig. 3a, b and Supplementary Fig. 4a), whereas other osteocyte-specific factors, including DMP1 and metalloendopeptidase homolog PEX (PHEX), were not altered (Fig. 3a). The reduced sclerostin expression resulted in activation of WNT/β-catenin signalling evidenced by increased nuclear accumulation of the non-phosphorylated, active form of β-catenin in osteocytes (Fig. 3c), and enhanced expression of WNT/β-catenin target genes including Axin-2 (*Axin2*), Dickkopf-4 (*Dkk-4*), lymphoid enhancer-binding factor 1 (*Lef-1*) and T-cell factor 1 (*Tcf-1*) (Fig. 3d). Of note, total cytoplasmic β-catenin levels were comparable between genotypes (Fig. 3c).

To decipher the molecular pathway upstream of decreased *Sost* transcription in *Phd2*$^{ot−}$ mice, we hypothesized that epigenetic regulation via SIRT1, a class III histone deacetylase (HDAC) could be involved. This assumption was based on two unrelated published observations showing that HIF signalling increases SIRT1 expression[32] and that SIRT1 directly and negatively regulates *Sost* expression in osteogenic cells in vivo[33,34]. However, a link between HIF-1α, SIRT1 and sclerostin has not been shown yet. We found that gene and protein expression of SIRT1 was increased in PHD2-deficient osteocytes in vivo (Fig. 3e and Supplementary Fig. 4b, c), whereas mRNA levels of other SIRT isoforms were not changed (Supplementary Fig. 4c). A similar effect was observed when the immortalized osteocyte cell line IDG-SW3 was treated with the PHD inhibitor IOX2 (IDG$^{IOX2}$), which was added to the culture medium when these cells started to express DMP1 during differentiation. This treatment increased HIF-1α and SIRT1 levels, accompanied by decreased mRNA and protein levels of sclerostin and increased WNT/β-catenin signalling (Fig. 3f and Supplementary Fig. 4d–i). Moreover, ChIP-qPCR analysis in IDG$^{IOX2}$ cells showed a significant enrichment of SIRT1 binding to specific regions in the *Sost* promoter (Supplementary Fig. 5a), and a marked decrease in histone 3 lysine 9 (H3K9) acetylation at these interrogated *Sost* promoter regions (highest at 998–1115 bp upstream of the start codon, region 1; Supplementary Fig. 5b), indicating epigenetic control of *Sost* expression. To prove a causative link between

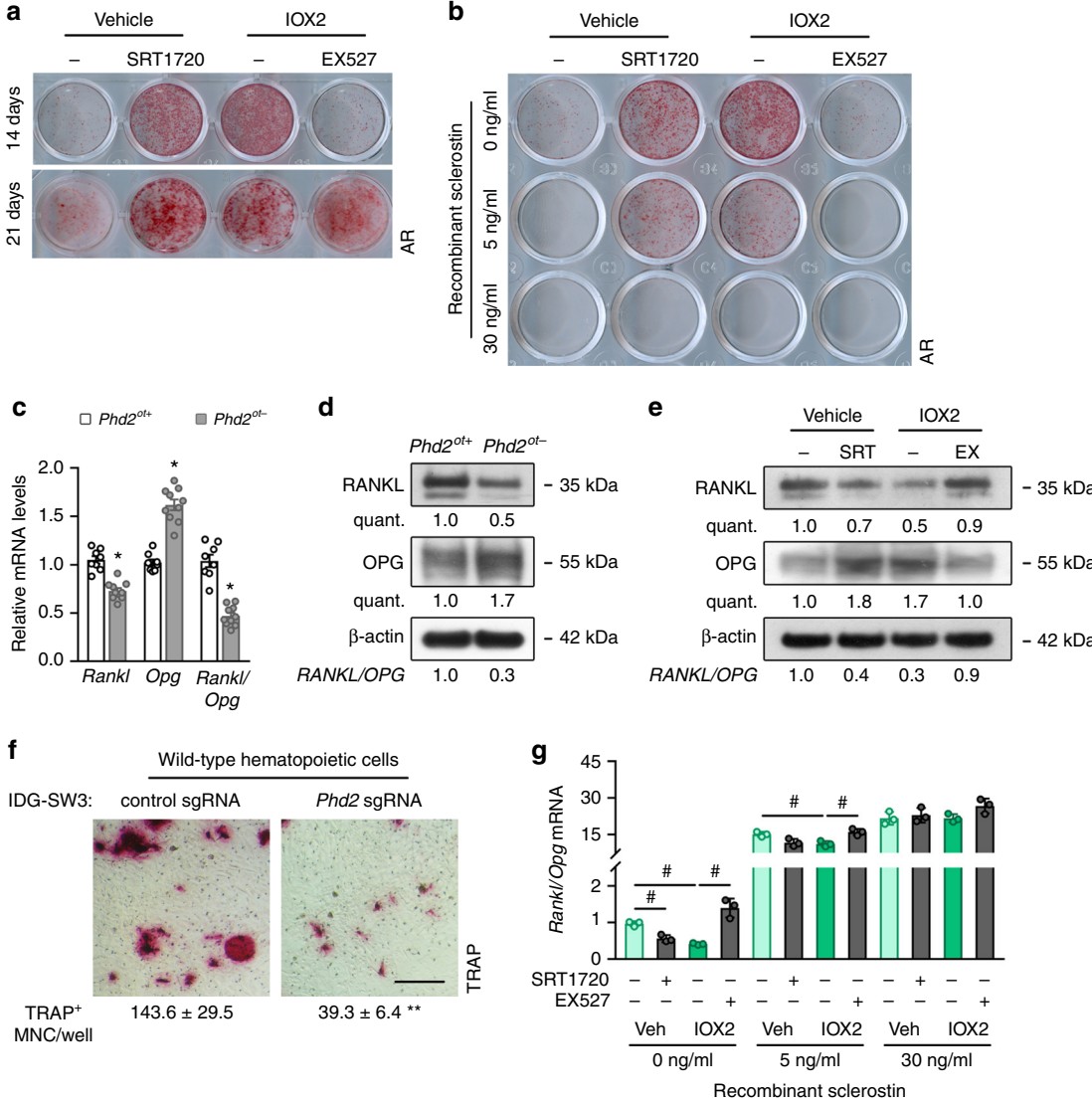

**Fig. 4** Decreased sclerostin affects osteoblasts and osteoclasts. **a** Alizarin Red staining of IDG^VEH or IDG^IOX2 cells after 14 or 21 days of osteogenic differentiation ($n = 3$). IDG^VEH cells were treated with vehicle (−) or SRT1720 (SRT); IDG^IOX2 cells were treated with vehicle (−) or EX527 (EX). **b** Alizarin Red staining of IDG^VEH or IDG^IOX2 cells after 14 days of osteogenic differentiation in the presence of recombinant sclerostin ($n = 3$). Cells were treated as in **a**. **c** Rankl and Opg mRNA levels in osteocyte-enriched bone fractions of 8-week-old mice ($n = 8$ Phd2^ot+–10 Phd2^ot-). **d** RANKL, OPG and β-actin immunoblot on whole-cell extracts isolated from osteocyte-enriched bone fractions. **e** RANKL, OPG and β-actin immunoblot on whole-cell extracts isolated from IDG^VEH or IDG^IOX2 cells after 14 days of osteogenic differentiation. Cells were treated as in **a**. **f** Quantification of TRAP-positive multinuclear cells (MNC), formed after co-culturing wild-type hematopoietic cells with IDG-SW3 cells, in which Phd2 was deleted or not via CRISPR-Cas9 ($n = 3$). **g** Rankl/Opg mRNA ratio in IDG^VEH or IDG^IOX2 cells after 14 days of osteogenic differentiation in the presence of recombinant sclerostin ($n = 3$). Cells were treated as in **a**. Data are means ± SEM. Immunoblot images are representative of three experiments. *$p < 0.05$ vs. Phd2^ot+ (Student's t-test), #$p < 0.05$ (two-way ANOVA)

HIF-SIRT1-*Sost*-β-catenin, we used two approaches. First, to substantiate that SIRT1 action mediated the effect of HIF signalling on *Sost* expression, we treated IDG^IOX2 cells with EX527, a potent and specific SIRT1 inhibitor. This treatment had no effect on SIRT1 levels, nor on SIRT1 binding to the *Sost* promoter, but caused a marked increase in H3K9 acetylation at the *Sost* promoter resulting in higher mRNA and protein levels of sclerostin (Fig. 3f and Supplementary Figs. 4d, e and 5c, d) and decreased WNT/β-catenin signalling (Fig. 3f and Supplementary Fig. 4f–i). Of note, culturing vehicle-treated IDG-SW3 cells (IDG^VEH) in the presence of the SIRT1 activator SRT1720 had opposite effects and underlined the role of SIRT1 activity in regulating *Sost* transcription (Fig. 3f and Supplementary Fig. 4d–i and 5c, d). Second, to exclude that activation of WNT/β-catenin

signalling resulted from direct SIRT1-mediated deacetylation of β-catenin[35], we added recombinant sclerostin, an approach that would only reverse the increased non-phosphorylated β-catenin levels when they were caused by decreased *Sost* expression. In line with this hypothesis, adding sclerostin to the culture medium of IDG^IOX2 or SRT1720-treated IDG^VEH cells inhibited the increase in active β-catenin levels (Fig. 3g). These data indicate that inhibition of PHDs in osteocytes increases SIRT1 expression, which modifies H3K9 acetylation of the *Sost* promoter and consequently suppresses its transcription, resulting in increased β-catenin activity.

**Decreased sclerostin affects osteoblasts and osteoclasts.** Next, we investigated the cellular effects of this enhanced WNT/β-

catenin signalling. IDG$^{IOX2}$ cells showed enhanced osteogenic differentiation capacity (Fig. 4a), as evidenced by Alizarin Red staining. Consistent with the above-described molecular pathway, the increase in osteogenic differentiation was blocked by inhibiting SIRT1 activity (Fig. 4a) or by adding recombinant sclerostin (Fig. 4b). Of note, treating IDG$^{VEH}$ cells with the SIRT activator SRT1720 increased osteogenic differentiation (Fig. 4a), which correlated with decreased sclerostin levels (Fig. 3f), and this effect was reversed by adding sclerostin (Fig. 4b).

Inhibition of WNT/β-catenin signalling by sclerostin can, besides reducing osteogenic differentiation, also stimulate osteoclastogenesis through modulation of the expression of the pro-osteoclastogenic protein receptor activator of nuclear factor kappa-B ligand (RANKL) and the RANKL decoy receptor osteoprotegerin (OPG)[13,14]. In agreement herewith, mRNA and protein expression analysis of the osteocyte-enriched bone fraction of $Phd2^{ot−}$ long bones or IDG$^{IOX2}$ cells showed a decrease in RANKL, whereas the expression of OPG was increased, resulting in a decreased ratio of RANKL/OPG (Fig. 4c–e). To link the decrease in RANKL/OPG ratio with impaired osteoclastogenesis in $Phd2^{ot−}$ mice, we co-cultured PHD2-deficient IDG-SW3 cells with osteoclast precursors from wild-type mice and treated them with 1α25-dihydroxyvitamin D$_3$ to stimulate osteoclast formation. Of note, PHD2-deficient IDG-SW3 cells display similar changes in HIF-1α-SIRT1-sclerostin signalling as IDG$^{IOX2}$ cells (Supplementary Fig. 6a-c). PHD2-deficient IDG-SW3 cells showed a reduced capacity to support osteoclastogenesis, consistent with the decrease in the RANKL/OPG ratio (Fig. 4f and Supplementary Fig. 6d, e). The decreased RANKL/OPG ratio in IDG$^{IOX2}$ cells was reversed by SIRT1 inhibition and we noted that SIRT1 activation is sufficient to reduce RANKL/OPG ratio in vehicle-treated cells, an effect that is rescued by adding sclerostin (Fig. 4e, g). Together, these data indicate that increased HIF signalling in osteocytes affects osteogenesis and osteoclastogenesis through a SIRT1-mediated decrease in sclerostin levels.

Lastly, we questioned whether the high bone mass phenotype observed in $Phd2^{ot−}$ mice was also attributed to increased SIRT1. Treatment of mutant mice with EX527 completely blunted the increase in trabecular bone volume (Fig. 5a, b and Supplementary Fig. 7a, b), and negatively affected cortical bone mass (Fig. 5a, c). Mechanistically, bone formation was significantly reduced, as evidenced by dynamic histomorphometric analysis, serum osteocalcin levels and $Ocn$ mRNA levels (Fig. 5d–f and Supplementary Fig. 7c). Moreover, EX527 treatment significantly increased the number of osteoclasts and $Rankl$/$Opg$ mRNA levels in mutant mice (Fig. 5g–i), indicative of increased bone resorption. These cellular effects were accompanied by high femoral sclerostin expression and low WNT/β-catenin signalling without affecting SIRT1 mRNA and protein levels (Fig. 5j–n and Supplementary Fig. 7d, e), analogous to the in vitro data. In addition, treatment of wild-type mice with SRT1720 resulted in a high-bone mass phenotype (Fig. 5a–i and Supplementary Fig. 7a–c), associated with low sclerostin expression and increased WNT/β-catenin signalling in bone (Fig. 5j–n and Supplementary Fig. 7e), thereby confirming our in vitro results and further supporting the reciprocal relationship between SIRT1 activity and sclerostin levels.

**HIF-1α mediates the changes in SIRT1.** The data discussed above indicate that PHD2 silencing in osteocytes results in activation of the hypoxia signalling pathway, with increased HIF-1α but not HIF-2α levels, thereby influencing WNT/β-catenin signalling. To prove that HIF activation was mediating this effect, we genetically silenced either HIF-1α or HIF-2α in IOX2-treated

IDG-SW3 cells using shRNA. Knockdown of HIF-1α significantly reduced SIRT1 levels in IDG$^{IOX2}$ cells, an effect that was associated with increased sclerostin levels (Fig. 6a, b) and impaired WNT/β-catenin signalling (Fig. 6b–f). In contrast, silencing HIF-2α in IDG$^{IOX2}$ cells did not affect SIRT1, sclerostin or WNT/β-catenin signalling (Fig. 6a–f). Moreover, hypoxic culture increased, in a HIF-1α-mediated way, osteocytic WNT/β-catenin signalling through a SIRT1-dependent decrease in sclerostin expression (Supplementary Fig. 8a–g and Supplementary Fig. 9a–g), thereby further underscoring the importance of PHD-mediated oxygen sensing by osteocytes to balance HIF levels.

**$Phd2^{ot−}$ mice show an enhanced skeletal angiogenic response.** Enhanced osteogenesis is often coupled with angiogenesis[22,24,36], possibly because the bone anabolic response requires sufficient supply of nutrients. CD31 immunostaining revealed a higher number of large blood vessels in the metaphysis and cortex of $Phd2^{ot−}$ long bones (Fig. 7a–d). Molecularly, mRNA levels of angiogenic factors like vascular endothelial growth factor ($Vegf$) and placental growth factor ($Plgf$) were upregulated in femurs of $Phd2^{ot−}$ mice, an effect that was recapitulated in cultured osteocytes (Fig. 7e). To test the functional significance of this increased production of angiogenic growth factors, we cultured human umbilical vein endothelial cells (HUVECs) in IDG-SW3-conditioned medium (Fig. 7f). Conditioned medium from cells with increased HIF signalling (IDG$^{IOX2}$ cells) increased the proliferation rate and the endothelial cell network formation of HUVECs compared to conditioned medium from IDG$^{VEH}$ cells (Fig. 7g–i). These beneficial effects were largely dependent on VEGF production by the osteocytes, as blocking VEGF signalling by addition of anti-VEGF$_{164}$ antibody or recombinant soluble VEGF receptor 1 (also known as fms-like tyrosine kinase-1 or sFlt-1) hampered the formation of network-like structures (Fig. 7h, i). This proangiogenic potential of IDG$^{IOX2}$ cells was HIF-dependent and isoform-specific, as silencing of HIF-1α, but not HIF-2α (Fig. 6a), in these cells reversed the effects on angiogenic growth factor production and endothelial cell behaviour (Fig. 7f–i and Supplementary Fig. 10a, b). Together, these data show that silencing of PHD2 in osteocytes promotes angiogenesis, through HIF-1α-dependent production of angiogenic growth factors.

Given the intimate connection between angiogenesis and osteogenesis in HIF-driven bone formation, we questioned whether enhanced SIRT1 signalling affects these processes accordingly. Treatment of $Phd2^{ot−}$ mice with the SIRT1 inhibitor EX527 completely blunted the increase in bone mass (Fig. 5a–c), but did not affect the increase in blood vessel number (Supplementary Fig. 10c, d). Accordingly, $Vegf$ mRNA levels remained elevated in femora collected from EX527-treated mutant mice (Supplementary Fig. 10e), presumably because HIF-1α signalling was still highly active (Fig. 5j). Similarly, activation of SIRT1 signalling in wild-type mice using SRT1720 resulted in a high-bone mass phenotype (Fig. 5a–c) without increasing bone vascularity or changing the production of VEGF (Supplementary Fig. 10c–e). In conclusion, these results indicate that enhanced SIRT1-dependent WNT/β-catenin signalling in $Phd2^{ot−}$ mice is the main driver of the anabolic effect on bone, which can be uncoupled from angiogenesis.

**$Phd2^{ot−}$ mice are protected from osteoporotic bone loss.** We finally tested whether the anabolic and anti-catabolic response in $Phd2^{ot−}$ mice can diminish bone loss in two osteoporotic models and whether the HIF-SIRT1-sclerostin pathway is involved. First, we subjected mice to hindlimb unloading, which reflects disuse-induced bone loss[37] and is characterized mainly by decreased

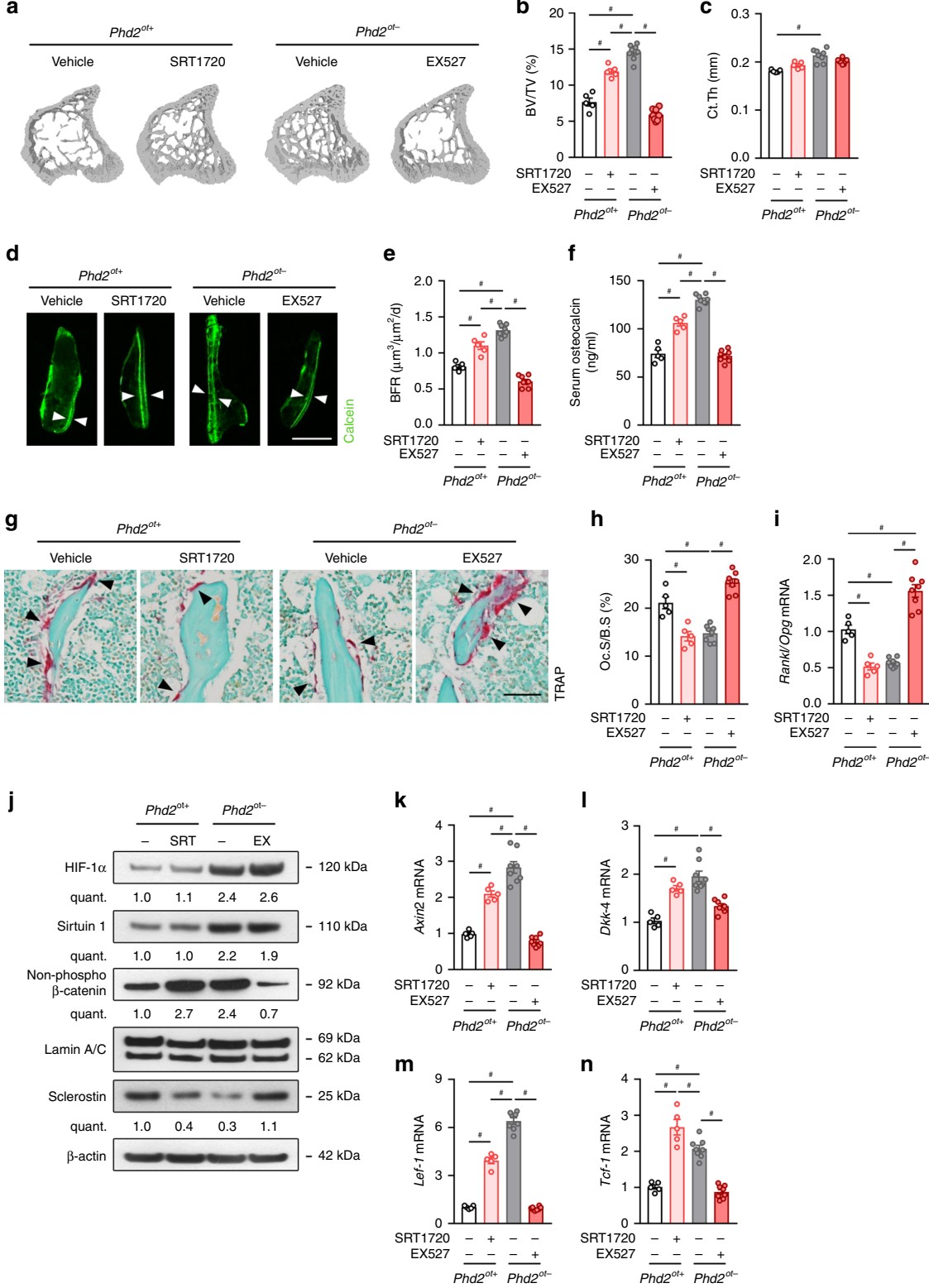

**Fig. 5** SIRT1 regulates bone mass in $Phd2^{ot-}$ mice. **a–c** 3D microCT models of the tibial metaphysis (**a**) and quantification of trabecular bone volume (BV/TV; **b**) and cortical thickness (Ct.Th; **c**) in 8-week-old mice ($n = 5$ $Phd2^{ot+}$–8 $Phd2^{ot-}$). Mice were treated with vehicle, SRT1720 or EX527 for 5 weeks. **d, e** Calcein labelling of trabecular mineralizing surfaces (**d**) with quantification (**e**) of the bone formation rate (BFR) ($n = 5$ $Phd2^{ot+}$–8 $Phd2^{ot-}$). White arrowheads in **d** indicate calcein incorporation. **f** Serum osteocalcin levels ($n = 5$ $Phd2^{ot+}$–8 $Phd2^{ot-}$). **g, h** TRAP staining (**g**) of the tibial metaphysis with quantification (**h**) of the osteoclast surface per bone surface (Oc.S/B.S) ($n = 5$ $Phd2^{ot+}$–8 $Phd2^{ot-}$). Black arrowheads in **g** indicate osteoclasts. **i** $Rankl/Opg$ mRNA levels in osteocyte-enriched bone fractions ($n = 5$ $Phd2^{ot+}$–8 $Phd2^{ot-}$). **j** HIF-1α, Sirtuin 1, non-phosphorylated (non-phospho) β-catenin, Lamin A/C (nuclear cell extracts), and sclerostin and β-actin immunoblot (whole-cell extracts). Protein extracts are isolated from osteocyte-enriched bone fractions. Results are representative of three experiments. **k–n** $Axin2$ (**k**), $Dkk-4$ (**l**), $Lef-1$ (**m**) and $Tcf-1$ (**n**) mRNA levels in osteocyte-enriched bone fractions ($n = 5$ $Phd2^{ot+}$–8 $Phd2^{ot-}$). Data are means ± SEM. $^{\#}p < 0.05$ (two-way ANOVA). Scale bars in **a** and **g** are 50 µm

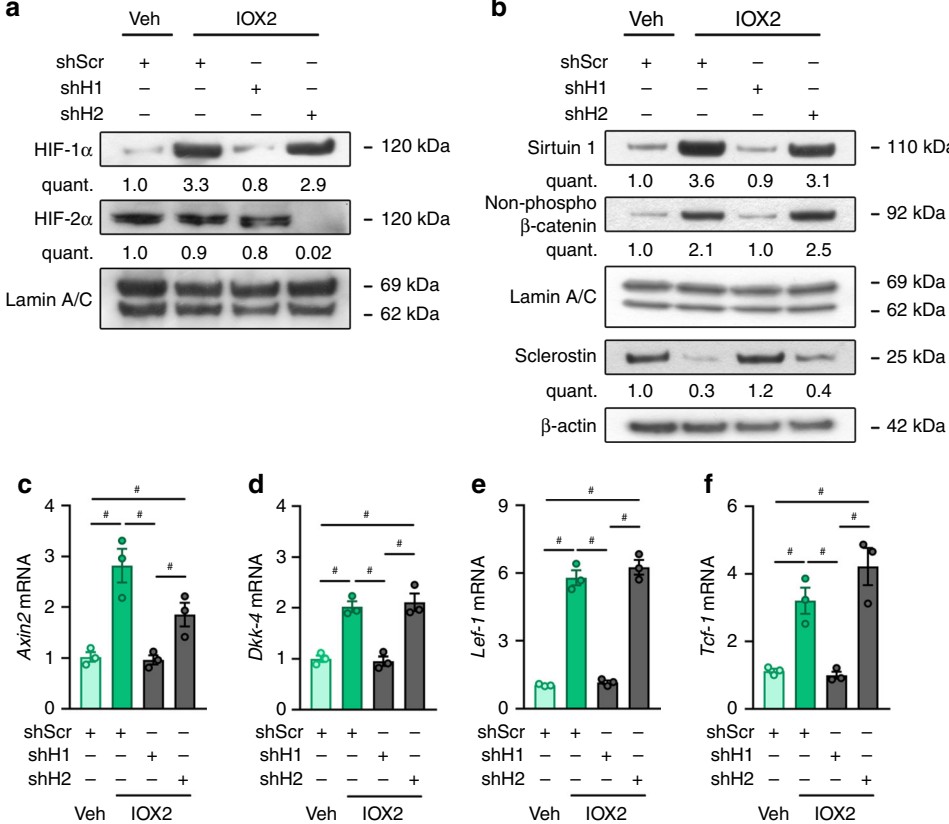

**Fig. 6** HIF-1α controls SIRT1 expression. **a** HIF-1α, HIF-2α and Lamin A/C immunoblot on nuclear cell extracts derived from IDG$^{VEH}$ or IDG$^{IOX2}$ cells after transduction with scrambled shRNA (shScr), shHIF-1α (shH1) or shHIF-2α (shH2). Results are representative of three experiments. **b** Sirtuin 1, non-phosphorylated (non-phospho) β-catenin, Lamin A/C (nuclear cell extracts), and sclerostin and β-actin immunoblot (whole-cell extracts). Protein extracts are derived from IDG$^{VEH}$ or IDG$^{IOX2}$ cells after genetic silencing of HIF-1α (shH1) or HIF-2α (shH2). Results are representative of three experiments. **c–f** *Axin2* (**c**), *Dkk-4* (**d**), *Lef-1* (**e**), *Tcf-1* (**f**) mRNA levels in IDG$^{VEH}$ or IDG$^{IOX2}$ cells after genetic silencing of HIF-1α (shH1) or HIF-2α (shH2) ($n = 3$). Data are means ± SEM. $^{\#}p < 0.05$ (one-way ANOVA)

bone formation together with increased bone resorption. Increased sclerostin levels likely contribute to this type of osteoporosis[11]. Hindlimb unloading through tail suspension of 16-week-old male *Phd2*$^{ot+}$ mice for 4 weeks resulted in marked trabecular and cortical bone loss (BV/TV: −36%, Ct.Th: −17%; Fig. 8a–c, Supplementary Fig. 11a, b and Supplementary Table 1). In contrast, unloaded *Phd2*$^{ot-}$ mice lost significantly less bone compared to grounded conditions (BV/TV: −26%, Ct.Th: −10%; Fig. 8a–c, Supplementary Fig. 11a, b and Supplementary Table 1). The reduction in bone mass in unloaded mice was caused by changes in bone formation and resorption in both genotypes, but was less pronounced in *Phd2*$^{ot-}$ animals: the bone formation rate was only moderately decreased in *Phd2*$^{ot-}$ animals (−65% in *Phd2*$^{ot+}$ vs −26% in *Phd2*$^{ot-}$; Fig. 8d–f and Supplementary Table 1) as were osteocalcin levels, and osteoclast number or *Rankl/Opg* ratio were not increased, whereas they were significantly upregulated in *Phd2*$^{ot+}$ mice (Fig. 8g–i and Supplementary Table 1).

Molecularly, unloading had no manifest effect on HIF-1α expression in either genotype, whereas SIRT1 mRNA and protein levels were significantly more decreased in unloaded *Phd2*$^{ot+}$ mice than in *Phd2*$^{ot-}$ mice (mRNA: −62% in *Phd2*$^{ot+}$ vs. −28% in *Phd2*$^{ot-}$, protein: −56% in *Phd2*$^{ot+}$ vs. −39% in *Phd2*$^{ot-}$; Fig. 8j and Supplementary Fig. 11c). This difference in SIRT1 levels was associated with a significant increase in sclerostin expression in unloaded *Phd2*$^{ot+}$ mice (mRNA: 8-fold, protein: 3.3-fold), but only a mild increase in *Phd2*$^{ot-}$ animals (mRNA:

3.8-fold, protein: 1.7-fold; Fig. 8j and Supplementary Fig. 11d). This resulted in impaired WNT/β-catenin signalling in unloaded *Phd2*$^{ot+}$ mice, evidenced by reduced nuclear accumulation of non-phosphorylated β-catenin and *Axin2*, *Dkk-4*, *Lef-1* and *Tcf-1* transcript levels (Fig. 8j–n and Supplementary Table 1), whereas WNT/β-catenin signalling remained higher in unloaded *Phd2*$^{ot-}$ mice compared to both grounded and unloaded wild-type animals (Fig. 8k–n and Supplementary Table 1).

In the second model, we studied whether osteocytic PHD2 deletion protects mice from oestrogen deficiency-induced osteoporotic bone loss, by performing ovariectomy (OVX) or sham operation on 10-week-old female mice. Similar to skeletal unloading, OVX in *Phd2*$^{ot+}$ mice resulted in a strong reduction in bone mass (BV/TV: −55%, Ct.Th: −11%), which was significantly less pronounced in *Phd2*$^{ot-}$ animals (BV/TV: −38%, Ct.Th: −5%; Fig. 9a–c, Supplementary Fig. 12a, b and Supplementary Table 2). The adverse effect of oestrogen deficiency on skeletal homoeostasis is mainly caused by an osteoclast-driven response[38] and, accordingly, we observed a significant increase in the number of osteoclasts after OVX, likely resulting from the elevated *Rankl/Opg* ratio (Fig. 9d–f and Supplementary Table 2). However, osteoclast number and *Rankl/ Opg* ratio were not or only marginally affected in *Phd2*$^{ot-}$ mice after OVX (Fig. 9d–f and Supplementary Table 2). Bone formation was not differentially changed by ovariectomy between the genotypes, evidenced by histomorphometric analysis after calcein labelling and serum osteocalcin levels (Fig. 9g–i and

Supplementary Table 2). Similar to the effect observed in the skeletal unloading model, PHD2 deletion in osteocytes prevented the molecular response of OVX on the HIF-1α-SIRT1-sclerostin-β-catenin pathway observed in $Phd2^{ot+}$ mice. Indeed, $Phd2^{ot-}$ mice showed no effect on HIF-1α, no decrease in SIRT1 mRNA and protein levels, no increase in sclerostin expression and

persistent WNT/β-catenin signalling (Fig. 9j–n, Supplementary Fig. 12c, d and Supplementary Table 2).

While the initiation of osteoporosis is angiogenesis-independent, accumulating evidence suggests that the local blood supply or decreased angiogenesis contributes, at least in part, to the bone loss[39,40]. In line with these observations, hindlimb unloading or ovariectomy of wild-type animals resulted in decreased

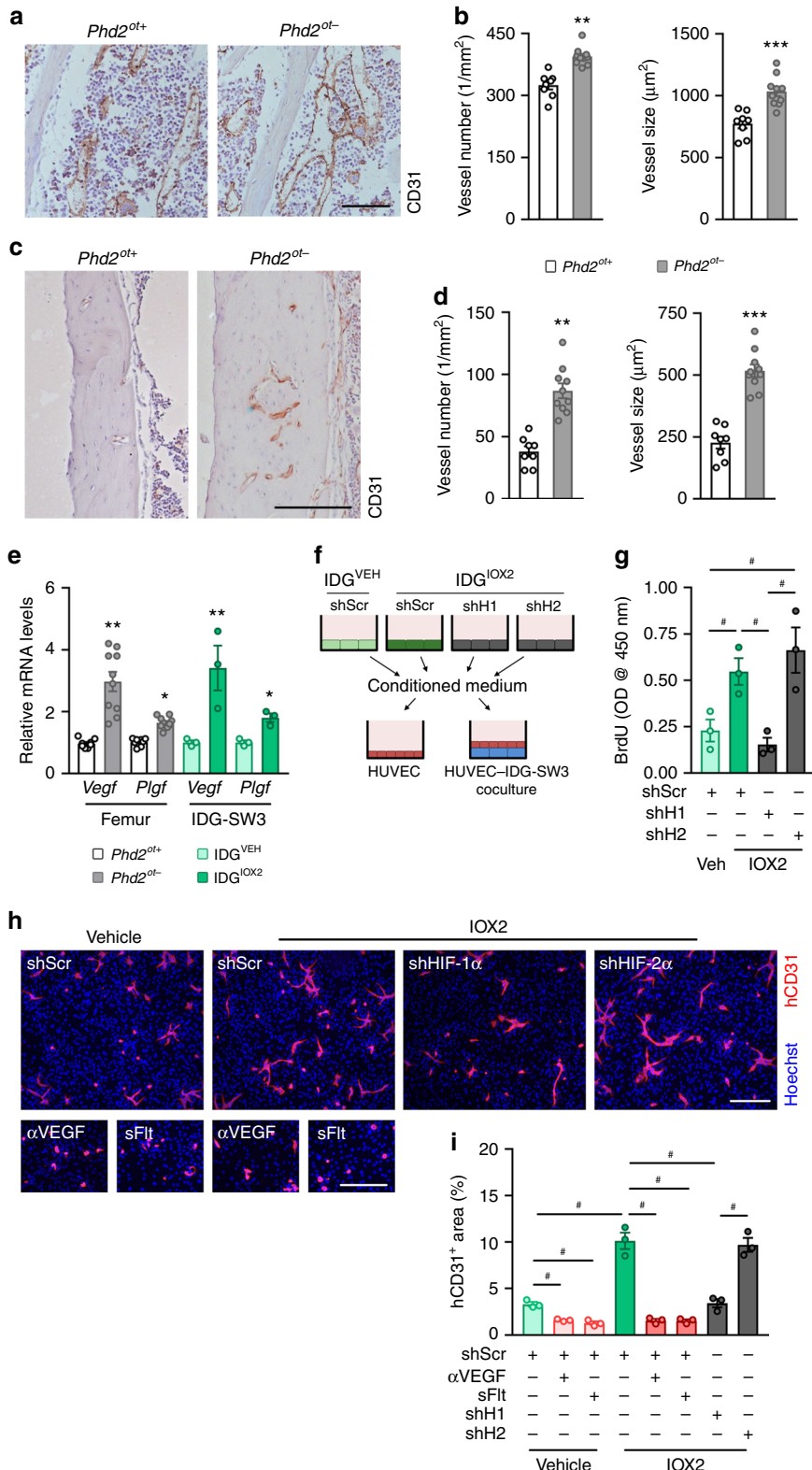

angiogenesis and impaired femoral *Vegf* mRNA expression (Supplementary Figs. 11e–g and 12e–g). In contrast, unloaded or ovariectomized *Phd2^ot−* mice did not display changes in vascularization compared to their respective controls, suggesting that elevated HIF-1α signalling was sufficient to maintain vascular density in these models (Supplementary Figs. 11e–g and 12e–g). Taken together, deletion of PHD2 in osteocytes protects mice from disuse and oestrogen deficiency-induced bone loss by preventing osteoclast-mediated bone resorption while sustaining bone formation and a pro-angiogenic response.

## Discussion

Osteocytes are crucial for the maintenance of a healthy skeleton by coordinating the function of osteoblasts and osteoclasts in response to hormonal and environmental stimuli, likely including oxygen fluctuations. Here, we show that the osteocytic oxygen sensor PHD2 controls postnatal bone homoeostasis by orchestrating the communication with osteoblasts and osteoclasts, through SIRT1-dependent epigenetic regulation of sclerostin expression.

The long lifespan of osteocytes together with their peculiar location within the bone matrix suggest that oxygen and nutrient delivery must be well controlled to ensure proper functioning. Our data indicate that the expression of the oxygen sensor PHD2 is higher in osteocytes compared to other skeletal cell types, suggesting that osteocytes are perfectly equipped to act as fast responders to changes in oxygen levels. However, we observed only few osteocytes with HIF-1α expression in wild-type animals, even though these cells are reported to reside in a low oxygen environment[16]. Likely, these relatively low oxygen levels are still sufficient to support PHD enzymatic activity, which is in line with other studies that report a low number of HIF-positive osteocytes in cortical bone[26]. On the other hand, PHD2 deletion in osteocytes gives a strong bone anabolic response and this phenotype clearly indicates that osteocytic oxygen sensing is an important regulator of bone homoeostasis by avoiding inappropriate HIF signalling. We want to highlight that deletion of the oxygen sensor PHD2 during bone development was the cause of increased HIF signalling and not an acute lack of oxygen. The active HIF signalling together with the increased vascular supply of oxygen and nutrients likely explains the bone anabolic effect in *Phd2^ot−* mice. This response may differ from severe ischaemic or hypoxic conditions, where nutrient and oxygen supply is limited and thereby probably cause activation of additional pathways, besides HIF signalling.

The expression of *Sost* is regulated by stimuli such as mechanical forces[11,41,42], growth factors[43] and hormones[44,45]. Our study now clearly identifies the local microenvironment, involving the sensing of oxygen levels, as an important modifier of *Sost* expression. More precisely, HIF-1α stabilization in osteocytes increases the expression and activity of SIRT1, a NAD$^+$-dependent HDAC, which can inhibit *Sost* expression[33].

Our in vivo data further underscore the importance of the HIF-1α-SIRT1-sclerostin connection as inhibition of SIRT1 activity in *Phd2^ot−* mice upregulates *Sost* levels manifestly and reverses the high bone mass phenotype, without affecting HIF-1α levels. These findings, together with similar in vitro observations, highlight that SIRT1 is downstream of HIF-1α signalling.

Consistent with the known role of sclerostin as inhibitor of canonical WNT signalling, we observed activation of the WNT/β-catenin pathway in PHD2-deficient osteocytes, evidenced by (i) nuclear accumulation of non-phosphorylated β-catenin, (ii) increased expression of downstream target genes, (iii) increased osteoblast-mediated bone formation and (iv) decreased osteoclast-dependent bone resorption through the modulation of RANKL and OPG. Direct regulation of OPG expression by HIF, as has recently been described[26] is less likely, because the increase in OPG levels in PHD2-deficient osteocytes is prevented when SIRT is inhibited. Our in vivo and in vitro findings further indicate that the molecular mechanism causing increased WNT/β-catenin signalling in PHD2-deficient osteocytes involves HIF-1α-mediated activation of SIRT1, which on its turn decreases sclerostin levels. The direct deacetylation of β-catenin by SIRT1, as was recently described in mesenchymal stem cells[35], is less probable because activation of β-catenin was reversed by adding recombinant sclerostin to IOX2-treated osteocytes, whereas SIRT1 levels remained high. Thus, we propose as model a PHD2—HIF-1α–SIRT1—sclerostin—WNT/β-catenin connection as response to changes in HIF levels in osteocytic lacunae.

Concomitant with the enhanced osteogenic response, osteocyte-specific deletion of PHD2 resulted in highly vascularized long bones, which is in line with the common notion that the hypoxia signalling pathway is a positive regulator of skeletal mass and angiogenesis[24,46]. Originally, increased bone vascular density has been proposed as the main mechanism responsible for HIF-mediated effects on bone formation[22], but recent reports indicate that HIF can regulate bone mass independently from changes in angiogenesis[25,26]. Our results expand this mechanistic model and show that PHD2/HIF-1α signalling in osteocytes also controls bone homoeostasis by orchestrating the communication with osteoblasts and osteoclasts and simultaneously adapting the vasculature, although the anabolic bone response was independent from the increase in angiogenesis.

The importance of WNT/β-catenin signalling for bone homoeostasis is underscored by the observation that several skeletal disorders, including osteoporosis, are associated with deregulation of this pathway[47]. Results from our present study and studies by others indicate that with ageing or after skeletal unloading or ovariectomy, mice show decreased bone mass due to reduced bone formation and increased bone resorption, an effect that is mediated by the manifest increase in osteocytic *Sost* expression and concomitant downregulation of WNT/β-catenin signalling[11,48,49]. Moreover, antibodies that neutralize sclerostin have shown their potential as treatment of osteoporotic bone loss in preclinical models and clinical trials[50–52]. Here, we

**Fig. 7** Deletion of PHD2 in osteocytes stimulates angiogenesis. **a, b** CD31 immunostaining (**a**) of the tibial metaphysis with quantification (**b**) of blood vessel number and size in 8-week-old mice ($n = 8$ *Phd2^ot+*–10 *Phd2^ot−*). **c, d** CD31 immunostaining (**c**) of the cortical diaphysis of tibiae with quantification (**d**) of blood vessel number and size ($n = 8$ *Phd2^ot+*–10 *Phd2^ot−*). **e** *Vegf* and *Plgf* mRNA levels in femora of 8-week-old mice ($n = 8$ *Phd2^ot+*–10 *Phd2^ot−*) and in vehicle (IDG$^{VEH}$) or IOX2-treated IDG-SW3 (IDG$^{IOX2}$) cells ($n = 3$). **f** Scheme of experimental design. HIF-1α (shH1) or HIF-2α (shH2) were silenced in IDG$^{IOX2}$ cells and a scrambled shRNA (shScr) was used as control. Conditioned medium was subsequently used to culture HUVECs in monoculture or in co-culture with IDG-SW3 cells. **g** Proliferation of HUVECs in monoculture according to the scheme in **f** ($n = 3$). **h, i** Human CD31 (hCD31) immunostaining (**h**) and quantification (**i**) of hCD31-positive surface in HUVECs co-cultured with IDG-SW3 cells, according to the scheme in **f**. When indicated, cells were treated with anti-VEGF$_{164}$ antibody or recombinant mouse soluble VEGFR-1 (sFlt) ($n = 3$). Data are means ± SEM. *$p < 0.05$ vs. *Phd2^ot+* or IDG$^{VEH}$, **$p < 0.01$ vs. *Phd2^ot+* or IDG$^{VEH}$, ***$p < 0.01$ vs. *Phd2^ot+* or IDG$^{VEH}$ (Student's *t*-test), #$p < 0.05$ (ANOVA: one-way ANOVA in **g**; two-way ANOVA in **i**). Scale bars in **a** and **c** are 100 μm, scale bar in **h** is 500 μm

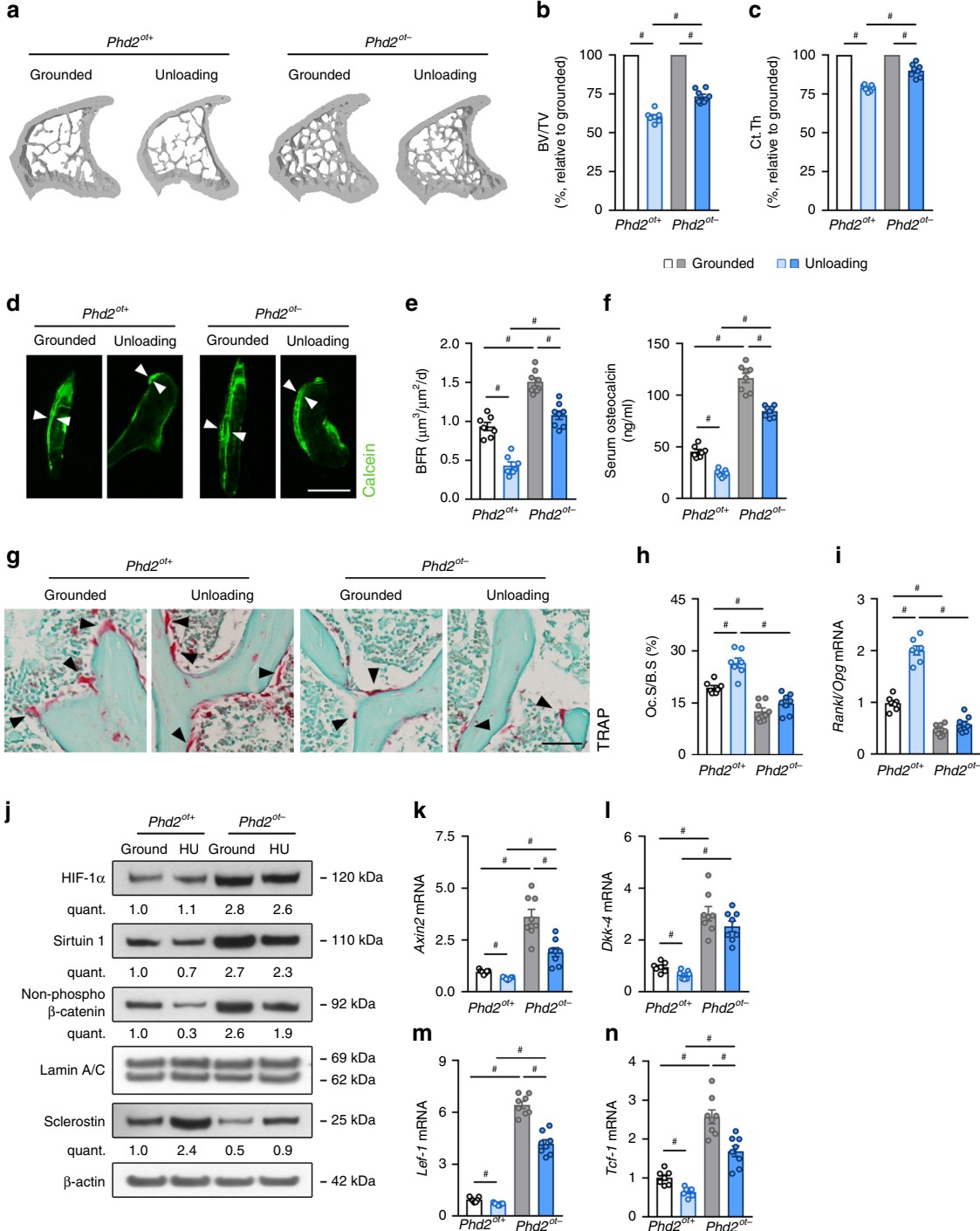

**Fig. 8** $Phd2^{ot-}$ mice are protected from disuse-induced bone loss. **a–c** 3D microCT models of the tibial metaphysis (**a**) and quantification of trabecular bone volume (BV/TV) (**b**) and cortical thickness (Ct.Th) (**c**) 4 weeks after hindlimb unloading and expressed as a percentage relative to grounded mice ($n = 7$ $Phd2^{ot+}$–8 $Phd2^{ot-}$). **d, e** Calcein labelling of trabecular mineralizing surfaces (**d**) with quantification (**e**) of the bone formation rate (BFR) ($n = 7$ $Phd2^{ot+}$–8 $Phd2^{ot-}$). White arrowheads in **d** indicate calcein incorporation. **f** Serum osteocalcin levels ($n = 7$ $Phd2^{ot+}$–8 $Phd2^{ot-}$). **g, h** TRAP staining (**g**) of the tibial metaphysis with quantification (**h**) of the osteoclast surface per bone surface (Oc.S/B.S) ($n = 7$ $Phd2^{ot+}$–8 $Phd2^{ot-}$). Black arrowheads in **g** indicate osteoclasts. **i** $Rankl/Opg$ mRNA levels in osteocyte-enriched bone fractions ($n = 7$ $Phd2^{ot+}$–8 $Phd2^{ot-}$). **j** HIF-1α, Sirtuin 1, non-phosphorylated (non-phospho) β-catenin, Lamin A/C (nuclear cell extracts), and sclerostin and β-actin immunoblot (whole-cell extracts). Protein extracts are isolated from osteocyte-enriched bone fractions. Results are representative of four experiments. HU is hindlimb unloading. **k–n** $Axin2$ (**k**), $Dkk-4$ (**l**), $Lef-1$ (**m**) and $Tcf-1$ (**n**) mRNA levels in osteocyte-enriched bone fractions ($n = 7$ $Phd2^{ot+}$–8 $Phd2^{ot-}$). Data are means ± SEM. $^{\#}p < 0.05$ (two-way ANOVA). Scale bars in **d** and **g** are 50 μm

demonstrate that selective deletion of PHD2 in osteocytes attenuates bone loss after skeletal unloading and ovariectomy, by preventing the increase in sclerostin levels and thus sustaining a WNT/β-catenin-driven anabolic response. In support of our work, administration of the pan-PHD inhibitor dimethyloxalylglycine rescued OVX-induced bone loss[39,40], although the efficacy of using these pan-PHD inhibitors for long-term treatment regimens has yet to be proven.

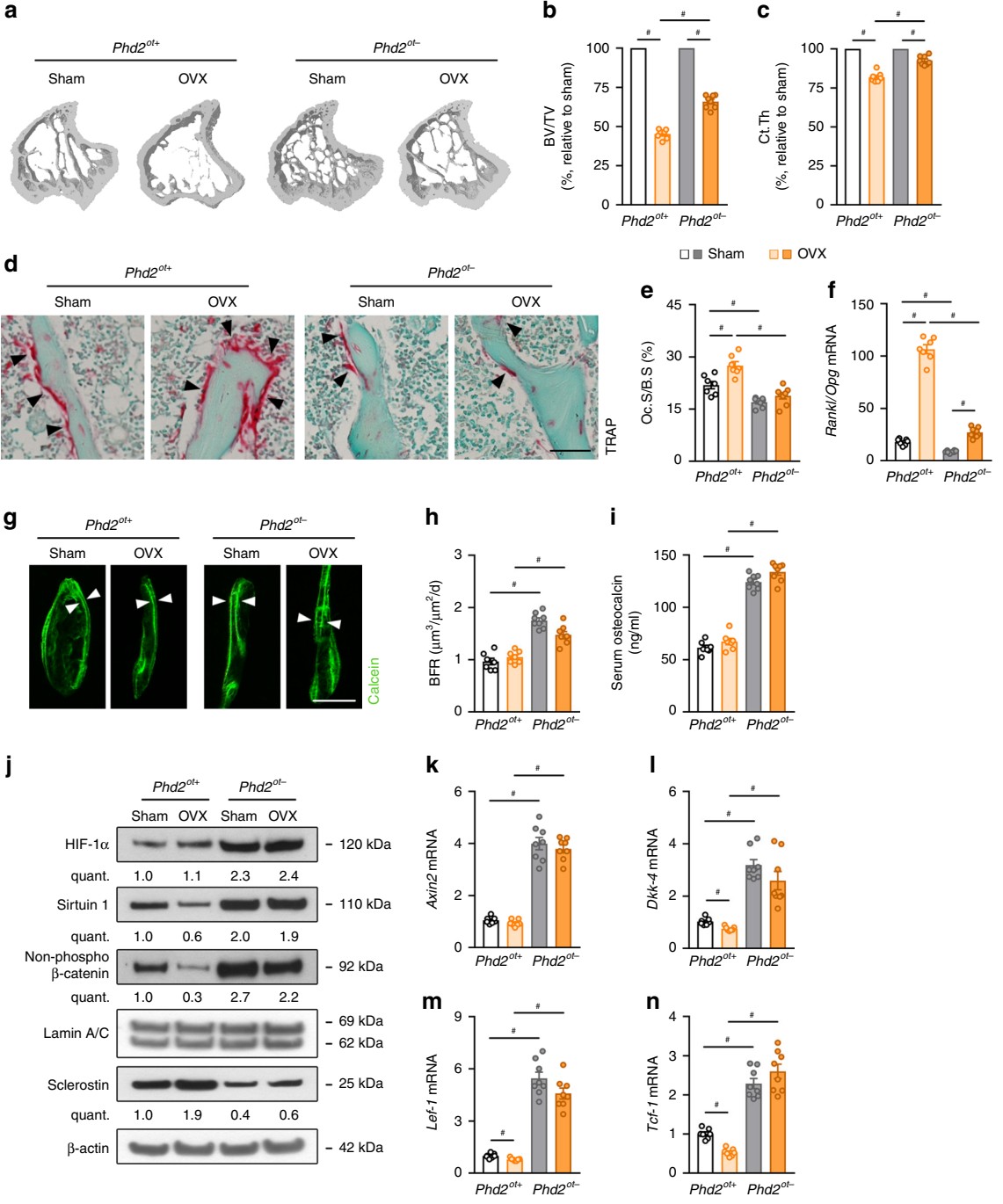

**Fig. 9** *Phd2^{ot−}* mice are protected from OVX-induced bone loss. **a–c** 3D microCT models of the tibial metaphysis (**a**) and quantification of trabecular bone volume (BV/TV) (**b**) and cortical thickness (Ct.Th) (**c**) 4 weeks after OVX and expressed as a percentage relative to sham-operated mice ($n = 7$ *Phd2^{ot+}*–8 *Phd2^{ot−}*). **d**, **e** TRAP staining (**d**) of the tibial metaphysis with quantification (**e**) of the osteoclast surface per bone surface (Oc.S/B.S) ($n = 7$ *Phd2^{ot+}*–8 *Phd2^{ot−}*). Black arrowheads in **e** indicate osteoclasts. **f** *Rankl/Opg* mRNA levels in osteocyte-enriched bone fractions ($n = 7$ *Phd2^{ot+}*–8 *Phd2^{ot−}*). **g**, **h** Calcein labelling of trabecular mineralizing surfaces (**g**) with quantification (**h**) of the bone formation rate (BFR) ($n = 7$ *Phd2^{ot+}*–8 *Phd2^{ot−}*). White arrowheads in **g** indicate calcein incorporation. **i** Serum osteocalcin levels ($n = 7$ *Phd2^{ot+}*–8 *Phd2^{ot−}*). **j** HIF-1α, Sirtuin 1, non-phosphorylated (non-phospho) β-catenin, Lamin A/C (nuclear cell extracts) and sclerostin and β-actin immunoblot (whole-cell extracts). Protein extracts are derived from osteocyte-enriched bone fractions. Results are representative of four experiments. **k–n** *Axin2* (**k**), *Dkk-4* (**l**), *Lef-1* (**m**) and *Tcf-1* (**n**) mRNA levels in osteocyte-enriched bone fractions ($n = 7$ *Phd2^{ot+}*–8 *Phd2^{ot−}*). Data are means ± SEM. $^{#}p < 0.05$ (two-way ANOVA). Scale bars in **d** and **g** are 50 μm

In conclusion, we have demonstrated that genetic ablation of PHD2 in osteocytes results in accumulation of bone mass by enhancing bone modelling in young and adult mice and reducing bone remodelling in aged mice. This high bone mass phenotype is elicited by a strong osteo-anabolic response associated with reduced bone resorption (Supplementary Fig. 13). Mechanistically, silencing PHD2 activates WNT/β-catenin signalling through SIRT1-dependent downregulation of sclerostin, thereby increasing osteoblast number and activity, while decreasing osteoclastogenesis and bone resorption. The enhanced osteogenic

response was coupled to increased angiogenesis, caused by HIF-1α-dependent stimulation of angiogenic growth factor production. Moreover, $Phd2^{ot-}$ mice were largely protected from bone loss caused by immobilization or oestrogen deficiency, by preventing the upregulation of sclerostin that is normally observed in these conditions. These data suggest that inhibiting PHD2 might be an appealing strategy to treat osteoporosis.

## Methods

**Animal models.** Osteocyte-specific deletion of PHD2 was obtained by crossing $Phd2^{fl/fl}$ mice[53] (provided by P. Carmeliet) with transgenic mice expressing Cre recombinase under the control of the *Dentin Matrix Protein 1* (*Dmp1*) gene promoter[54] (*Dmp1-Cre*[+] $Phd2^{fl/fl}$, referred to as $Phd2^{ot-}$). *Dmp1-Cre*[-] $Phd2^{fl/fl}$ (referred to as $Phd2^{ot+}$) littermates were used as control in all experiments; analysis was performed on 8-week-old male mice unless stated otherwise. All mice were on a 100% C57BL/6 background. To assess the impact of ageing on bone mass, male (age: 3 weeks, 8 weeks, 16 weeks, 20 weeks, 50 weeks) and female mice (age: 3 weeks, 8 weeks, 10 weeks, 14 weeks, 50 weeks) were used. SRT1720 (50 mg/kg body weight; Merck) or EX527 (2 mg/kg body weight; Sigma-Aldrich) was administered every other day via intraperitoneal injection from weaning (3 weeks of age) on until 8 weeks of age. At the time of sacrifice, mice were euthanized, blood was collected and long bones were dissected for RNA isolation, histology and microCT analysis. Mice were bred in conventional conditions in our animal housing facility (Proefdierencentrum Leuven, Belgium). Housing and experimental procedures were approved by the Institutional Animal Care and Research Advisory Committee of the KU Leuven (ethical approval number P214/2013).

Hindlimb unloading was achieved in 16-week-old male mice by a previously described tail suspension model[37]. Briefly, a flexible iron wire surrounded with orthopaedic tape was applied laterally along the tail of the mice. The mice were then positioned in a 40–45° head-down tilt, in order that the hindlimbs were lifted off the ground but the mice were still able to move around by using their forelimbs. As a control, pair-fed animals without tail suspension were used. Four weeks after hindlimb unloading, mice were sacrificed, blood was collected and long bones were dissected for RNA isolation, histology and microCT analysis.

Ovariectomy (OVX) was performed in 10-week-old female mice. After anaesthetization (pentobarbital, 60 mg/kg body weight), the left and right ovary were exposed and removed; in the sham-operated (sham) animals, the ovaries were exposed but left intact. Four weeks after OVX, mice were sacrificed, blood was collected and long bones were dissected for RNA isolation, histology and microCT analysis.

**MicroCT.** MicroCT analysis of the tibiae was performed ex vivo using the high resolution SkyScan 1172 system (Bruker; 50 kV, 200 µA, 0.5 mm aluminium filter) and in vivo using the SkyScan 1076 system (Bruker; 50 kV, 100 µA, 1 mm aluminium filter)[55]. Serial tomographs, reconstructed from raw data using the cone-beam reconstruction software (NRecon; Bruker), were used to compute trabecular and cortical parameters, respectively from the metaphyseal and mid-diaphyseal area. All measurements were performed according to the guidelines of the American Society for Bone and Mineral Research[56].

**Histology and (immuno)histochemical staining.** For H&E, tartrate-resistant acid phosphatase (TRAP), osteoid and CD31 staining, bones were fixed overnight at 4 °C in 2% paraformaldehyde, decalcified for 2 weeks in 0.5 M EDTA pH 7.5, and embedded in paraffin. Undecalcified bones (Von Kossa, calcein labelling) were fixed overnight at 4 °C in Burckhart's solution and embedded in methylmetacrylate (MMA).

Osteoblast parameters were measured on H&E-stained sections. Briefly, paraffin sections were stained with haematoxylin (Prosan) for 2 min, followed by staining with eosin (Richard-Alan) for 5 min. Osteoclasts were visualized on paraffin sections reacted for TRAP activity[57] and counterstained with Light Green SF Yellowfish (Merck). Osteoid was visualized on MMA sections by Goldner staining. Briefly, sections were stained with a mixture of 0.1% Ponceau de Xylidine (BDH), 0.1% fuchsine (BDH) and 0.2% Orange G (BDH) and counterstained with Light Green SF Yellowfish. For analysis of mineralized bone tissue, MMA sections were stained according to Von Kossa. Here, sections were treated with 5% AgNO3 for 1 h, resulting in silver deposits at sites of calcium incorporation. The staining was fixed with 5% Na2S2O3 and sections were counterstained with haematoxylin.

Immunohistochemical staining conditions were slightly adapted according to the antibody used. Generally, paraffin sections were de-waxed, rehydrated, incubated with Antigen Retrieval Solution (Dako) and washed in TBS. Endogenous peroxidase activity was blocked by immersing the sections in 0.3% H2O2 in methanol for 20 min. Unspecific antibody binding was blocked by incubating the sections in 2% BSA-supplemented TBS for 30 min. Subsequently, sections were incubated overnight with primary antibody against CD31 (550274, 1/50 dilution; BD Biosciences), HIF-1α (NB100–449, 1/50 dilution; Bio-Techne), sclerostin (AF1589, 1/200 dilution; Bio-Techne) or SIRT1 (07-131, 1/25 dilution; Merck). For visualization through a biotin-mediated reaction (CD31), slides were exposed to HRP-conjugated streptavidin (PerkinElmer) for 30 min and antibody binding was

envisaged by HRP activity on the colour substrate diaminobenzidine. Afterwards, slides were counterstained with haematoxylin. For HIF-1α, sclerostin and SIRT1 immunostaining, signal visualization was obtained using appropriate Cy3-labelled secondary antibodies. Hoechst staining was used to visualize cell nuclei.

**Histomorphometry.** Histomorphometric analysis of murine long bones was performed on an AxioPlan 2 microscope (Zeiss) with AxioVision software (Zeiss)[58]. Measurements were done on three sections, each at least 40 µm apart and data were presented according to the guidelines of the American Society for Bone and Mineral Research standardized histomorphometry nomenclature[59]. Briefly, osteoblast and osteocyte number and surface were quantified on H&E-stained sections; the number of viable osteocytes was determined by calculating the difference between total and empty lacunae. Osteoid, osteoblast and osteoclast surface were expressed relatively to the total bone surface. Trabecular bone volume was quantified in the secondary spongiosa of Von Kossa-stained sections. To analyse dynamic bone parameters, calcein (16 mg/kg body weight; Sigma-Aldrich) was administered via intraperitoneal injection 4 days and 1 day prior to sacrifice. The mineralizing surface (MS), mineral apposition rate (MAR) and bone formation rate (BFR) were quantified on unstained MMA-sections. MAR was quantified as the mean distance between all double fluorochrome labels, divided by the number of days between calcein injections. BFR was calculated as MS = (dLS + sLS/2)/BS, where dLS and sLS represent the bone surface covered by double (d) and single (s) calcein labels. Vascular density in the metaphysis was measured on CD31-stained sections and calculated relative to the total bone surface.

**Bone mechanical properties.** The biomechanical properties of femurs collected from 8-week-old male mice were determined by three-point bending analysis on a Bose ElectroForce 3100 system (Bose). Prior to testing, all bones were kept moist in gauze swabs soaked in PBS. Femora were loaded to failure in the anterior–posterior direction with a span length of 7 mm at a constant displacement rate of 0.5 mm/s. WinTest software was used to collect the load-displacement data at 250 data points per second for a total of 10 s. Work-to-failure and bone stiffness were determined from the applied load and displacement data.

**Whole blood analysis and serum biochemistry.** Whole blood collected at the time of sacrifice was either directly analysed for quantification of RBC number and haematocrit levels (University Hospital Leuven, Belgium), or centrifuged ($7300 \times g$ for 8 min) and the serum retained. Serum osteocalcin was measured by an in-house radioimmunoassay[60], serum collagen type I cross-linked C-telopeptide levels were measured by a RatLaps ELISA kit (Immunodiagnostic Systems) according to the manufacturer's instructions.

**IDG-SW3 cell culture.** IDG-SW3 cells (mycoplasma-tested), kindly provided by L. Bonewald[61], were expanded at permissive conditions (33 °C in nucleoside-supplemented αMEM with 10% FBS, 100 units/ml penicillin, 50 µg/ml streptomycin and 50 U/ml IFN-γ) on rat-tail collagen type I-coated plates (0.15 mg/ml collagen in 0.02 M acetic acid)[61]. IFN-γ was from Thermo Fisher Scientific, rat-tail collagen type I from BD Biosciences and other reagents were from Gibco. To inhibit PHDs[27], IDG-SW3 cells were treated with 10 µM IOX2 (IDG$^{IOX2}$; Sigma-Aldrich) or DMSO (1/5000 dilution; Sigma-Aldrich) as control (vehicle, IDG$^{VEH}$). After 48 h, cells were used for further experiments.

For osteogenic differentiation, IDG-SW3 cells were expanded and upon confluency, medium was switched to osteogenic differentiation medium, consisting of growth medium supplemented with 50 µg/ml ascorbic acid and 4 mM β-glycerophosphate (both Sigma-Aldrich) and cultured at 37 °C. After 7 days of osteogenic induction, when *Dmp1* gene expression was upregulated[61], cells were treated with either DMSO (1/5000 dilution) or IOX2 (10 µM). Two days later, IDG$^{VEH}$ cells were treated with 1 µM SRT1720 and IDG$^{IOX2}$ cells with 5 µM EX527, supplemented or not with recombinant human sclerostin (5 ng/ml or 30 ng/ml; Bio-Techne) for the following 5 or 12 days. Finally, proteins were extracted or mineral deposition was visualized using Alizarin Red, as described before[62,63]. To silence HIF-1α or HIF-2α in IDG-SW3 cells, we transduced these cells with a lentivirus carrying a shRNA against HIF-1α or HIF-2α (shHIF-1α and shHIF-2α, kindly provided by P. Carmeliet; MOI 10) in the presence of 8 µg/ml polybrene (Sigma-Aldrich)[27,30]. A lentivirus carrying a nonsense scrambled shRNA sequence (shScr) was used as a negative control (MOI 10). After 24 h, virus-containing medium was changed to normal culture medium and 48 h later, cells and conditioned medium were used for further experiments. To silence PHD2 in IDG-SW3 cells using CRISPR-Cas9, we transduced these cells with a lentivirus carrying a plasmid containing the Cas9 enzyme (lentiCRISPR v2; Addgene) and a sgRNA against *Phd2* (5′-GCCTGGGTAACAAGCAACCA-3′). A nonsense scrambled sgRNA (5′-GCTGATCTATCGCGGTCGTC-3′) was used as a negative control. After 24 h, virus-containing medium was changed to normal culture medium and cells were selected with puromycin (0.3 µg/ml) for 7 days before they were used in subsequent experiments.

To investigate the interaction of osteocytes with osteoclasts, bone marrow cells (collected after flushing long bones) were plated overnight in αMEM without nucleosides containing 10% FCS, 100 units/ml penicillin and 50 µg/ml streptomycin (all from Gibco). Non-adherent cells were collected and seeded onto

IDG-SW3 cells (i.e. day 0). Cells were treated with 1α25-dihydroxyvitamin D$_3$ ($10^{-8}$ M; Sigma-Aldrich) and medium was replaced after 3 days. At day 7, TRAP staining was performed and the number of TRAP-positive cells with more than three nuclei was counted as osteoclasts.

A co-culture system was used to study the interaction of osteocytes and endothelial cells[64]. Briefly, IDG-SW3 cells were seeded and 24 h later human umbilical cord vein endothelial cells (Lonza) were added. Cells were cultured for 7 days in conditioned medium derived from IDG$^{VEH}$ or IDG$^{IOX2}$ cells after transduction with shScr, shHIF-1α or shHIF-2α, with or without the addition of recombinant soluble fms-related tyrosine kinase-1 (sFlt-1; 100 ng/ml; Bio-Techne) or anti-murine VEGF-164 antibody (αVEGF$_{164}$; 200 ng/ml; Bio-Techne). After culture, the endothelial cells were visualized by staining with a mouse-anti-human CD31 primary antibody (1/1000; Dako) and the TSA Cyanine 3 System (PerkinElmer).

To assess osteocyte behaviour in hypoxia, IDG-SW3 cells were cultured in 1% oxygen (hypoxic glove box; Coy Lab Products). Briefly, after 7 days of osteogenic induction (growth medium supplemented with 50 μg/ml ascorbic acid and 4 mM β-glycerophosphate; both Sigma-Aldrich), cells were placed in the hypoxic glove box for 48 h in the presence of DMSO (1/5000 dilution) or EX527 (5 μM). Alternatively, prior to hypoxic culture, cells were transduced with a lentivirus carrying a shRNA against HIF-1α or HIF-2α (both MOI 10). A lentivirus carrying a nonsense scrambled shRNA sequence was used as a negative control (MOI 10).

**Primary cell isolation and culture**. Periosteal cells were isolated from long bones of 7–9-week-old male mice[64]. After dissection of muscle and connective tissue, epiphyses were embedded in 5% low melting point agarose (Lonza). Subsequently, periosteal cells were isolated by a twofold collagenase-dispase digest (3 mg/ml collagenase and 4 mg/ml dispase in αMEM with 2 mM glutaMAX$^{TM}$−1, containing 100 units/ml penicillin and 50 μg/ml streptomycin; all from Gibco). Cells obtained after 10 min of digest were discarded. The cells obtained during the second digest (50 min) were passed through a 70 μm nylon mesh, washed and directly used for RNA or protein analysis (see below).

For isolation of trabecular osteoblasts and bone marrow cells[62], long bones of 7–9-week-old male mice were cleaned thoroughly to remove muscle, connective tissue and periosteum. Subsequently, bones were incubated in a collagenase-dispase mixture (3 mg/ml collagenase and 4 mg/ml dispase in αMEM with 2 mM glutaMAX$^{TM}$-1, containing 100 units/ml penicillin and 50 μg/ml streptomycin; all from Gibco) for 20 min at 37 °C to remove remaining periosteal cells. Next, epiphyses were cut away and bone marrow was flushed out, which was collected after centrifugation (300 × g for 7 min) and used for subsequent analysis. The remaining bone was cut into small pieces and trabecular osteoblasts were isolated by incubating the fragments with the collagenase-dispase mixture for 30 min at 37 °C. Cells were passed through a 70 μm nylon mesh (BD Falcon), washed twice and used for RNA and protein analysis.

The osteocyte-enriched bone fraction was obtained after removal of the bone marrow, and sequential collagenase and EDTA digestions of the long bones[62]. Tibiae and femurs of 7–9-week-old mice were cleaned to remove muscle and connective tissue. Subsequently, bones were incubated in 2 mg/ml collagenase (in αMEM with 2 mM glutaMAX$^{TM}$-1, containing 100 units/ml penicillin and 50 μg/ml streptomycin; all from Gibco) for 30 min at 37 °C. Epiphyses were cut, bone marrow was flushed and bone was cut into smaller pieces. These fragments were incubated in 1 mg/ml collagenase mixture for 40 min at 37 °C and this cell suspension was discarded. The remaining bone chips were washed with PBS and incubated for 40 min at 37 °C with 5 mM EDTA in PBS. Cell suspension was again discarded and the bone fragments were finally incubated with 1 mg/ml collagenase mixture for 50 min at 37 °C. Cells were collected, passed through a 70 μm nylon mesh and washed twice. This osteocyte-enriched bone fraction was directly lysed for RNA and protein analysis.

For in vitro osteoclast formation[58], bone marrow cells (collected after flushing long bones) were plated overnight in αMEM without nucleosides containing 10% FCS, 100 units/ml penicillin and 50 μg/ml streptomycin (all from Gibco). Non-adherent cells were collected and plated in αMEM supplemented with 20 ng/ml macrophage colony-stimulating factor (M-CSF; Bio-Techne) and 100 ng/ml receptor activator of nuclear factor kappa-B ligand (RANKL; Peprotech) (i.e. day 1). At day 6, the number of TRAP-positive cells with more than three nuclei was counted as osteoclasts.

**Quantitative real-time (qRT)-PCR**. RNA isolation and qRT-PCR analysis were performed as described before[27,63]. RNA was collected and purified with the RNeasy Mini Kit (QIAGEN) according to the manufacturer's instructions. cDNA was synthesized from 1 μg RNA with reverse transcriptase Superscript II RT (Thermo Fisher Scientific). Gene expression was analysed by Taqman quantitative RT-PCR using custom-made primers and probes, or commercial primer sets (Integrated DNA Technologies, Inc.) as listed in Supplementary Table 3. Expression levels were normalized relative to the expression of *Hprt*. For quantification of gene expression, ΔΔCt method was used.

**Protein analysis by western blot**. For whole-cell lysates[27,63], cells were rinsed with ice-cold PBS and lysed in a total cell lysis buffer (50 mM Tris-HCl pH 8.5, 150 nM NaCl, 0.1% SDS, 1% NP40, 1% sodium desoxycholate, supplemented with protease

inhibitor mix consisting of 1 mM PMSF, 5 μg/ml aprotinine, 5 μg/ml leupeptin and 0.33 μg/ml antipain). Nuclear protein fractions[27,63] were prepared by lysing the cells first in a hypotonic buffer (20 mM Hepes pH 7.9, 10 mM KCl, 1.5 mM MgCl$_2$, 1 mM EDTA, 0.5% NP40, 1 mM DTT, supplemented with 1 mM Na$_3$VO$_4$, 20 mM NaF and protease inhibitor mix) for 15 min at 4 °C followed by mechanical disruption of the cell membranes. Nuclei were pelleted from the lysates by centrifugation (17,900 × g for 1 min at 4 °C). The pellet was resuspended in a nuclear extraction buffer (50 mM Hepes pH 7.9, 500 mM NaCl, 1% NP40, supplemented with protease inhibitor mix) and after sonication incubated for 15 min at 4 °C. Protein concentrations were determined with the BCA Protein Assay Reagent (Thermo Fisher Scientific).

Proteins were separated by SDS-PAGE under reducing conditions and transferred to a nitrocellulose membrane (GE Healthcare). Membranes were blocked with 5% dry milk or bovine serum albumin (Sigma-Aldrich) in Tris-buffered saline with 0.1% Tween20 for 60 min at room temperature and incubated overnight at 4 °C with primary antibodies. For detection of PHD2 (NB100−2219, 1/1000 dilution; Bio-Techne), sclerostin (AF1589, 1/1000 dilution; Bio-Techne), total β-catenin (D10A8, 1/1000 dilution; Cell Signaling Technology), RANKL (NB100-56388, 1/1000 dilution; Bio-Techne), OPG (AF459, 1/1000 dilution; Bio-Techne) and β-actin expression (A5441, 1/10000 dilution; Sigma-Aldrich), whole-cell lysates were used; for HIF-1α (NB100-105, 1/500 dilution; Bio-Techne), HIF-2α (ab199, 1/500 dilution; Abcam), SIRT1 (07-131, 1/1000 dilution; Merck), non-phosphorylated β-catenin (D13A1, 1/1000 dilution; Cell Signaling Technology) and Lamin A/C (sc-376248, 1/5000 dilution; Santa Cruz Biotechnology) detection, nuclear protein extracts were used. Bound primary antibody was visualized by enhanced chemiluminescence (PerkinElmer) after incubation with a species-specific HRP-conjugated secondary antibody (Dako). Uncropped scans are provided in Supplementary Fig. 14.

**Cell proliferation**. Cell proliferation was measured by 5′-bromo-2′-deoxyuridine (BrdU) incorporation, added during the last 4 h and detected using the Cell Proliferation Biotrack ELISA system (GE Healthcare). The results were normalized to the amount of DNA.

**Chromatin immunoprecipitation qPCR (ChIP-qPCR) analysis**. For ChIP-qPCR analysis[27], IDG-SW3 cells were first expanded and upon confluency, medium was switched to osteogenic differentiation medium and cells were further cultured at 37 °C. After 7 days of osteogenic induction, cells were treated with either DMSO or IOX2 (10 μM). Two days later, IDG$^{VEH}$ cells were in addition treated with 1 μM SRT1720 and IDG$^{IOX2}$ cells were treated with 5 μM EX527 for the following 5 days. Cultured cells were fixed using 1% formaldehyde, washed and collected by centrifugation (1000 × g for 5 min at 4 °C). The pellet was resuspended in RIPA buffer (50 mM Tris-HCl pH 8, 150 mM NaCl, 2 mM EDTA, 1% Triton-X100, 0.5% sodium deoxycholate, 1% SDS, 1% protease inhibitors), homogenized, incubated on ice for 10 min and sonicated. The samples were centrifuged (16,000 × g for 10 min at 4 °C) and from the supernatant, shared chromatin was used as input and immunoprecipitation was performed with anti-SIRT1 antibody (07-131; Merck) or anti-histone H3 acetyl lysine 9 (H3K9Ac) antibody (39137; Active Motif). After precipitation using Pierce Protein A/G Magnetic Beads (Thermo Fisher Scientific), followed by RNA and protein digestion, DNA was purified using Agencourt AMPure XP (Beckman Coulter) according to the manufacturer's instructions. RT-qPCR was performed using SYBR® GreenER™ qPCR SuperMix Universal (Thermo Fisher Scientific) and primers for specific *Sost* promoter regions (Supplementary Table 4).

**Statistics**. Data are presented as means ± SEM. *n* values represent the number of independent experiments performed or the number of individual mice phenotyped. For each independent in vitro experiment, at least three technical replicates were analysed. For immunoblots, representative images are shown of at least three independent experiments using samples from different mice/cell lysates. No statistical tests were used to pre-determine sample size for in vitro and in vivo studies, and no data was excluded from this study. No statistical methods were used for randomization for in vitro and in vivo experiments, but investigators were blinded to group allocation. Data were analysed by two-sided two-sample Student's *t*-test, and one-way or two-way ANOVA with Tukey–Kramer post-hoc test using the NCSS statistical software. Variance was similar between tested groups and differences were considered statistically significant at $p < 0.05$.

**Data availability**. All data are available from the corresponding author upon reasonable request.

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

## Acknowledgements

We are grateful to Dr. Lynda Bonewald for providing the IDG-SW3 cell line and Dr. Massimiliano Mazzone for access to the hypoxic glove box. We wish to thank Sophie Torrekens, Riet Van Looveren and Erik Van Herck for excellent technical assistance. G. C. acknowledges funding support from the Research Foundation—Flanders (FWO: G.0A72.13, G.096414 and G0A4216N) and P.C. from long-term structural funding—Methusalem Funding by the Flemish Government. S.S. is a postdoctoral fellow from the FWO (12H5917N).

## Author contributions

S.S. and G.C. designed research, S.S., I.S., K.M. and B.T. performed research, P.H.M. and P.C. contributed new reagents, S.S., I.S. and K.M. analysed the data; and S.S. and G.C. wrote the manuscript.

## Additional information

**Competing interests:** The authors declare no competing interests.

