## [Peer Review File · Nature Communications]

Reviewers' comments:

Reviewer #1 (Remarks to the Author):

In this interesting paper the authors show that deletion of the oxygen sensor PHD2 (possibly expressed at high levels in osteocytes) targeted to osteocytes (and late osteoblasts), leading to stabilization of HIF1a in these cells, increases bone mass via an increase in bone formation and a decrease in bone resorption. The authors then show that this led to an increased Sirtuin 1-dependent deacetylation of the Sost promoter and a decrease in sclerostin expression, explaining at least in part the phenotype. They also looked at the effects of this deletion and of pharmacological regulators of PHD or Sirtuin 1 on bone changes and on angiogenesis. The results confirmed the proposed mechanism for bone but showed that the regulation of angiogenesis is, at least in part, independent. Silencing of PHD2 promotes angiogenesis through HIF1a-dependent production of angiogenic factors but this is independent of SIRT1. They then move to test the effects of this deletion on two models of bone loss, OVX and unloading. Although they conclude that PHD2 deletion prevented bone loss in these two models the results are far from convincing. Overall an interesting paper but one that will require significant revisions to reach publication level.

Critique:

The main strength of this paper is the convincing demonstration of a pathway linking oxygen sensing to sclerostin regulation and thereby bone formation and, to some degree bone resorption. The main weaknesses are 1/ The lack of in situ osteocyte analysis (lacunar size by BSEM, in situ expression by hybridization and/or immunocytochemistry); 2/ The lack of convincing data on the two clinically relevant models; 3/ A lack of clarity on the results and interpretation of two way ANOVA results in these studies; 4/ A lack of integration of the findings in a model of remodeling (how can low oxygen lead to decrease resorption when haversian remodeling is induced by lack of central vascularization?) and how can it increase bone formation, an effect that may decrease access to oxygen sources from osteocytes.

Specific points:

- 1- Are osteocytes really in a low oxygen environment or is this an assumption? Only one paper makes this claim and this appears a weak demonstration in my view. Some osteocytes (mid cortex) might indeed be, but not all osteocytes? If this is true anyway it should lead to increased angiogenesis and remodeling to allow vessels penetration. Is that what is happening? Or is there a contradiction in the observed effects? Please clarify and discuss this important matter.
- 2- The changes in unloading and OVX must be expressed as % change and tested as such. From looking at the data, this reviewer failed to see a convincing "protection" in the KO animals: they start from much higher values and of course the values remain higher after OVX or unloading. But is it from "protection" or simply from starting higher? The authors need to recalculate the data as percent changes if possible. The 2 way ANOVA could resolve the question if it was presented: is there interaction between the two effects, OVX/unloading and KO? Looking at the graphs one can see the exact same trends in the KO group, strongly suggesting lack of protective effects.
- 3- The authors need to perform in situ analysis of osteocytes to confirm changes in gene expression and their homogeneous (or not) distribution. HIF1a, RANKL, OPG, Sost...
- 4- To conclude at lack of cell-autonomous changes (although unexpected) in osteoclasts would require mix-and-match experiments with osteoblasts or better with osteocytes.
- 5- RANKL OPG data should not be in supplemental 2f but in the main paper.
- 6- An important technical concern is that since the paper addresses the effects of oxygen sensing, one wonders whether the primary cells are affected by the isolation and culture procedures. How do the authors prevent changes in oxygen sensing when isolating osteocytes?
- 7- If the authors insist on saying that "selective PHD2 inhibition might be beneficial to prevent bone loss..." (end of discussion) they have to tell the reader how selectivity could be achieved. In my view not an easy task!
- 8- An effort needs to be done in the discussion to present an integrative view of how it all works and how these data explain (or not) the mechanisms of bone remodeling, particularly in the cortex.

Reviewer #2 (Remarks to the Author):

Authors explore the role of the oxygen sensing pathway in osteocytes and its consequences on bone remodelling using a number of cell and animal models. This study provides interesting novel findings, builds an interesting mechanistic circuit and thus adds new information relevant to the field of bone homeostasis.

The study provides good evidence for the involvement of PHD2 in bone homeostasis. However, there is no direct proof for a role of hypoxia itself. Since "off-target" effects are always difficult to exclude, I think that authors should explore the direct effects of hypoxia on the molecular pathways mentioned in the study. This could be done in some of the animal or cell culture models used.

Additionally, they should provide, as supplementary data, much more details about the various methods and techniques used, so that other researchers can reproduce/refute their results. In the figures, it is often difficult to know how many independent experiments and how many replicates in a single experiment were used.

We thank the reviewers for their positive feedback and constructive comments. We have addressed all major concerns and performed a series of additional *in vitro* and *in vivo* experiments to complete the study according to the reviewers' suggestions. These new data have been included and discussed in the revised manuscript. For clarity, we refer to the new figure panels in our answers.

Reviewer #1 (Remarks to the Author):

In this interesting paper the authors show that deletion of the oxygen sensor PHD2 (possibly expressed at high levels in osteocytes) targeted to osteocytes (and late osteoblasts), leading to stabilization of HIF1a in these cells, increases bone mass via an increase in bone formation and a decrease in bone resorption. The authors then show that this led to an increased Sirtuin 1-dependent deacetylation of the Sost promoter and a decrease in sclerostin expression, explaining at least in part the phenotype. They also looked at the effects of this deletion and of pharmacological regulators of PHD or Sirtuin 1 on bone changes and on angiogenesis. The results confirmed the proposed mechanism for bone but showed that the regulation of angiogenesis is, at least in part, independent. Silencing of PHD2 promotes angiogenesis through HIF1a-dependent production of angiogenic factors but this is independent of SIRT1. They then move to test the effects of this deletion on two models of bone loss,

OVX and unloading. Although they conclude that PHD2 deletion prevented bone loss in these two models the results are far from convincing. Overall an interesting paper but one that will require significant revisions to reach publication level.

Critique:

The main strength of this paper is the convincing demonstration of a pathway linking oxygen sensing to sclerostin regulation and thereby bone formation and, to some degree bone resorption.

The main weaknesses are 1/ The lack of *in situ* osteocyte analysis (lacunar size by BSEM, *in situ* expression by hybridization and/or immunocytochemistry); 2/ The lack of convincing data on the two clinically relevant models; 3/ A lack of clarity on the results and interpretation of two way ANOVA results in these studies; 4/ A lack of integration of the findings in a model of remodeling (how can low oxygen lead to decrease resorption when haversian remodeling is induced by lack of central vascularization?) and how can it increase bone formation, an effect that may decrease access to oxygen sources for osteocytes.

We thank the reviewer for the positive and helpful comments, and hope that these revisions help clarify the remaining concerns.

Specific points:

1- Are osteocyte really in a low oxygen environment or is this an assumption? Only one paper makes this claim and this appears a weak demonstration in my view. Some osteocytes (mid cortex) might indeed be, but not all osteocytes? If this is true anyway it should lead to increased angiogenesis and remodeling to allow vessels penetration. Is that what is happening? Or is there a contradiction in the observed effects? Please clarify and discuss this important matter.

We appreciate the reviewer's concerns on whether osteocytes are truly in a low oxygen environment. Indeed, one recent study suggests that osteocytes in the cortical bone reside in a low oxygen environment, characterized by an oxygen tension of ± 30 mmHg (or ± 4 % O₂) (Spencer, JA Nature 2014). These low oxygen levels are reported to decrease PHD enzymatic activity *in vitro* (Jiang, BH Am. J. Physiol. 1996 and Epstein, AC Cell 2001), but are likely sufficient to maintain osteocytic PHD function *in vivo*. Indeed, *in situ* analysis of osteocytic HIF-1 α expression in wild type

animals revealed a low number of HIF-1 α -positive cells (revised Supplementary Figure 1d), which is in line with previous reports (Wu, C Genes & Development 2015). We now discuss our findings, together with earlier studies, as part of the discussion in the revised manuscript (p. 16):

“Our data indicate that the expression of the oxygen sensor PHD2 is higher in osteocytes compared to other skeletal cell types, suggesting that osteocytes are perfectly equipped to act as fast responders to changes in oxygen levels. However, we observed only few osteocytes with HIF-1 α expression in wild type animals, even though these cells are reported to reside in a low oxygen environment. Likely, these relatively low oxygen levels are still sufficient to support PHD enzymatic activity, which is in line with other studies that report a low number of HIF-positive osteocytes in cortical bone. On the other hand, PHD2 deletion in osteocytes gives a strong bone anabolic response and thus clearly indicates that osteocytic oxygen sensing is an important regulator of bone homeostasis by avoiding inappropriate HIF signalling.”

For correct understanding, we want to stress that activation of the hypoxia signalling pathway in our mouse model is caused by genetic deletion of PHD2 in osteocytes and not by affecting oxygen availability, because we aimed to investigate the role of the oxygen sensors in bone homeostasis. The observed bone anabolic response in *Phd2*^{ot-} mice is in line with previous findings in mouse models with active HIF signalling in osteolineage cells, obtained by *Vhl* deletion or expression of stable HIF-1 α (Wang, Y JCI 2007; Regan, JN PNAS 2014; Weng, T JBMR 2014; Stegen, S Cell Metab 2016). In none of these models, nor in *Phd2*^{ot-} mice, cortical bone remodelling was observed (Figure 1 for Reviewer). Likely, the developmental models of HIF activation that result in a high bone mass phenotype differ from acute (pathological) hypoxic or ischemic conditions, which may probably initiate a bone remodelling response. In our study, the phenotype is rather caused by progressive cortical bone mass modelling associated with increased cortical vessel density (revised Figure 7c,d). Whether acute or chronic oxygen scarcity affects cortical bone remodelling through modulation of the HIF-1 α – SIRT1 – sclerostin signalling pathway remains to be determined, but is outside the scope of this study.

Figure 1. Lack of cortical bone remodelling in *Phd2*^{ot-} mice. CD31 immunostaining (left panels) and TRAP staining (right panels) of cortical bone and tibial metaphysis of 8-week-old male *Phd2*^{ot-} mice. Note that blood vessels are present in the cortical bone (red arrows), whereas TRAP-positive osteoclasts (yellow arrow) are located on the bony trabeculae in the metaphysis but not in the cortex. Scale bar is 100 μ m.

To clarify this important issue, we adapted the discussion of the revised manuscript as follows (p. 16):

*“We want to highlight that deleting the oxygen sensor PHD2 during development was the cause of increased HIF signalling and not an acute lack of oxygen. The active HIF signalling together with the increased vascular supply of oxygen and nutrients likely explains the bone anabolic effect in *Phd2*^{ot-}”*

mice. This response may differ from hypoxic or ischemic conditions, where nutrient and oxygen supply is limited and thereby probably cause activation of additional pathways, besides HIF signalling.”

Together, these findings further support our claim that PHD2-mediated oxygen sensing is critical to avoid undesirable HIF-1 α accumulation in osteocytes, which would otherwise result in a high bone mass phenotype. Therefore, we adapted the title of our manuscript to “*Osteocytic oxygen sensing controls bone mass through epigenetic regulation of sclerostin*” to better reflect our findings.

2- The changes in unloading and OVX must be expressed as % change and tested as such. From looking at the data, this reviewer failed to see a convincing "protection" in the KO animals: they start from much higher values and of course the values remain higher after OVX or unloading. But is it from "protection" or simply from starting higher? The authors need to recalculate the data as percent changes if possible. The 2 way ANOVA could resolve the question if it was presented: is there interaction between the two effects, OVX/unloading and KO? Looking at the graphs one can see the exact same trends in the KO group, strongly suggesting lack of protective effects.

We acknowledge that the statistical evaluation of the data of the bone pathology models was insufficiently described, as was correctly pointed out by the reviewer, and it was therefore unclear for the reader that the data were already analysed by two-way ANOVA in the original manuscript.

To increase clarity, we added supplementary tables to describe the main effects and interaction terms (Supplementary Table 1 and 2). This analysis shows a significant main effect of genotype, with a significantly higher % change from baseline in *Phd2*^{0t-} mice compared to *Phd2*^{0t+}. The main effect for treatment is also significant, with unloading or OVX significantly increasing the % change from baseline. In addition, the interaction effect (genotype*treatment) is significant, indicating that the effect of unloading or OVX was significantly greater in *Phd2*^{0t+} than in *Phd2*^{0t-} mice.

For further clarification, we have now plotted the % change in BV/TV and Ct.Th, relative to grounded/sham-operated mice and included this information in the revised Figure 8b,c and Figure 9b,c.

3- The authors need to perform in situ analysis of osteocytes to confirm changes in gene expression and their homogeneous (or not) distribution. HIF1a, RANKL, OPG, Sost...

We fully understand the reviewer’s concern that changes in gene expression do not always translate in alterations at the protein level. However, we want to point out that the expression of HIF-1 α , SIRT1 and sclerostin was already analysed by Western blot on osteocytes directly isolated from cortical bone, an approach that is considered to be highly quantitative. We have now added RANKL and OPG protein levels and these data prove that deletion of PHD2 decreases RANKL and increases OPG levels *in vivo* (revised Figure 4d), which is in line with the reduced osteoclast number and activity *in vivo*.

In addition, we have now performed immunohistochemical stainings for HIF-1 α , SIRT1 and sclerostin to validate *in situ* our results obtained by Western blot. In wild type mice, we detected only few HIF-1 α -positive osteocytes (\pm 10%), located primarily in the central region of the cortical bone (revised Supplementary Figure 1d), which corresponds well with the relatively low HIF-1 α levels in the osteocyte-enriched bone fractions (revised Figure 1b). In accordance with the low HIF-1 α levels, we observed only few SIRT1-positive osteocytes, which was associated with high sclerostin expression (revised Supplementary Figures 4a,b). In contrast, deletion of PHD2 resulted in stabilization of HIF-1 α in the majority (\pm 75%) of the cortical osteocytes (revised Supplementary Figure 1d), a high number of SIRT1-positive cells and low sclerostin expression (revised

Supplementary Figure 4a,b). These findings highly correlate with our observations in the osteocyte-enriched bone fractions.

4- To conclude at lack of cell-autonomous changes (although unexpected) in osteoclasts would require mix-and-match experiments with osteoblasts or better with osteocytes.

In response to this question, we generated PHD2-deficient IDG-SW3 cells using CRISPR-Cas9 (revised Supplementary Figure 6a-c) and co-cultured these cells or control cells (scrambled sgRNA) with hematopoietic osteoclast precursor cells isolated from wild type mice. Upon stimulation with 1α 25-dihydroxyvitamin D₃, control IDG-SW3 cells adequately supported osteoclastogenesis, whereas PHD2-deficient IDG-SW3 cells failed to support the formation of TRAP⁺ multinuclear cells (revised Figure 4f). Mechanistically, PHD2-deficient IDG-SW3 cells showed a decrease in RANKL, whereas the expression of OPG was increased, resulting in a decreased RANKL/OPG ratio (revised Supplementary Figure 6d,e). These data are now described in the revised manuscript on page 10.

These results, together with the observation that osteoclast precursors from *Phd2*^{ot-} mice differentiate normally (revised Supplementary Figure 3d,e), indicate that osteocytic PHD2 regulates osteoclastogenesis in a paracrine manner.

5- RANKL OPG data should not be in supplemental 2f but in the main paper.

As requested, we have included the *Rankl* – *Opg* gene expression data in the revised Figure 4. In addition, we performed Western blot analysis for RANKL-OPG on osteocytes isolated from *Phd2*^{ot+} and *Phd2*^{ot-} mice and on IOX2-treated IDG-SW3 cells *in vitro*. Genetic or pharmacological inactivation of PHD2 confirmed the *Rankl* – *Opg* gene expression results, showing a decrease in RANKL and an increase in OPG levels thereby decreasing the RANKL/OPG ratio (revised Figure 4c-e).

6- An important technical concern is that since the paper addresses the effects of oxygen sensing, one wonders whether the primary cells are affected by the isolation and culture procedures. How do the authors prevent changes in oxygen sensing when isolating osteocytes?

We understand the concern of the reviewer that isolation and culture of primary osteocytes may affect their behaviour. However, we would first like to point out that all the analyses performed on osteocytes isolated from transgenic mice were achieved without cell culture, as collected cells were directly lysed in the respective buffers for RNA or protein isolation. In addition, HIF-1 α levels in osteocyte-enriched bone fractions isolated from wild type mice were relatively low (revised Figures 1b, 5j, 8j, 9j), which corresponds with the *in situ* analysis of HIF-1 α expression (revised Supplementary Figure 1d) and thereby indicates that the used procedure does not manifestly affect the obtained results.

For clarity, we described thoroughly the technical details of osteocyte isolation from bone and subsequent RNA/protein analysis in the Supplementary Methods section (p. 22-23).

7-If the authors insist on saying that "selective PHD2 inhibition might be beneficial to prevent bone loss..." (end of discussion) they have to tell the reader how selectivity could be achieved. It my view not an easy task!

We regret to have caused confusion regarding the therapeutic potential of targeting PHD2 to prevent bone loss, as we aimed to discuss selective inhibition of the PHD2 isoform and not osteocyte-specific targeting. In order to avoid misinterpretation, we removed this statement from the discussion in the revised manuscript.

8- An effort needs to be done in the discussion to present an integrative view of how it all works and how these data explain (or not) the mechanisms of bone remodeling, particularly in the cortex.

We refer to our answer to question 1 and we have, as suggested, summarized our findings in the revised Supplementary Figure 13. Moreover, we have expanded the discussion on page 19 as follows: *“In conclusion, we have demonstrated that genetic ablation of PHD2 in osteocytes results in accumulation of bone mass by enhancing bone modelling in young and adult mice and reducing bone remodelling in aged mice. This high bone mass phenotype is elicited by a strong osteo-anabolic response associated with reduced bone resorption (Supplementary Figure 13). Mechanistically, silencing PHD2 activates WNT/ β -catenin signalling through SIRT1-dependent downregulation of sclerostin, thereby increasing osteoblast number and activity, while decreasing osteoclastogenesis and bone resorption. The enhanced osteogenic response was coupled to increased angiogenesis, caused by HIF-1 α -dependent stimulation of angiogenic growth factor production. Moreover, Phd2^{ot}-mice were largely protected from bone loss caused by immobilization or oestrogen deficiency, by preventing the upregulation of sclerostin that is normally observed in these conditions. These data suggest that inhibiting PHD2 might be an appealing strategy to treat osteoporosis.”*

Reviewer #2 (Remarks to the Author):

Authors explore the role of the oxygen sensing pathway in osteocytes and its consequences on bone remodelling using a number of cell and animal models. This study provides interesting novel findings, builds an interesting mechanistic circuit and thus adds new information relevant to the field of bone homeostasis.

We would like to thank the reviewer for the assessment of our manuscript and the recognition that our proposed concept is relevant to the field of bone biology.

The study provides good evidence for the involvement of PHD2 in bone homeostasis. However, there is no direct proof for a role of hypoxia itself. Since "off-target" effects are always difficult to exclude, I think that authors should explore the direct effects of hypoxia on the molecular pathways mentioned in the study. This could be done in some of the animal or cell culture models used.

As suggested by the reviewer, we cultured IDG-SW3 cells in 1% oxygen and assessed the expression of HIF-1 α , SIRT1, sclerostin and non-phosphorylated β -catenin as a marker for active WNT signalling. As described in the revised Supplementary Figure 8, hypoxic culture resulted in (i) an increase in SIRT1, (ii) a decrease in sclerostin and (iii) activation of WNT/ β -catenin signalling in a HIF-1 α -dependent manner, thereby mirroring the changes observed in osteocytes isolated from *Phd2*^{ot-} mice and in IOX2-treated IDG-SW3 cells.

Moreover, we could demonstrate that the effects of hypoxia on sclerostin levels and WNT/ β -catenin signalling were caused by an increase in SIRT1, as treatment with the SIRT1-inhibitor EX527 normalized the hypoxia-induced effects to the level observed in normoxic control cells (revised Supplementary Figure 9).

Together, these data demonstrate that hypoxic osteocytes regulate WNT/ β -catenin signalling through a HIF-1 α – SIRT1 – sclerostin signalling axis, thereby mirroring the effects observed in osteocytes with genetic or pharmacological PHD2 inactivation. Whether acute or chronic hypoxia affects bone mass *in vivo* through a similar mechanism remains to be determined.

Additionally, they should provide, as supplementary data, much more details about the various methods and techniques used, so that other researchers can reproduce/refute their results.

As requested, we now extensively describe the used methods and techniques in the Methods and Supplementary Methods section.

In the figures, it is often difficult to know how many independent experiments and how many replicates in a single experiment were used.

We apologize for the insufficient information on the number of experimental replicates. For *in vivo* experiments, "n" indicates the number of animals used; for *in vitro* experiments, "n" indicates the number of independent experiments (for each independent experiments, at least 3 technical replicates per condition were used). We have included this information in the Methods section (p. 24-25).

Reviewer #3 (Remarks to the Author):

Several investigators have shown that HIF1 regulates bone mass, largely through altering activity of osteoprogenitors and osteoblasts. However, the mechanism by which this occurs is not defined. The purpose of this work was to determine whether prolyl hydroxylases are involved in this regulatory pathway. Secondly, constant remodeling of bone is required to maintain bone strength. Remodeling requires communication between osteoblasts and osteoclasts. The WNT/Beta-catenin antagonist, sclerostin, is an important mediator of the communication. However, factors that regulate sclerostin are undefined. The rationale for believing that HIF-1 is involved in this biology is the recognition that lacunae which contain osteocytes are often hypoxic.

This extremely well performed piece of research identifies PHD2 as the central mediator of bone mass. The pathway by which this happens is via PHD2 regulation of Sirtuin 1. A series of elegant genetically engineered mice and thorough in vitro studies have uncovered this biology, which suggests a totally new pathway for preventing bone loss, as occurs with osteoporosis, for example. Key findings include the observation that mice that are knocked out for PHD2 in osteocytes are protected from bone loss induced by both disuse and estrogen deficiency.

I looked for weaknesses in the study, but every time I thought of something they should do, they did it in the next paragraph. It is rare to find a paper where the statistical design and analyses are clearly stipulated and conducted. Nevertheless, I do have some high level concerns that should be addressed.

We thank the reviewer for the positive evaluation of our work.

1- The quantification of bone mineral density is done mainly using histology to measure trabecular bone volume and cortical thickness. It would be of interest to understand mechanical strength of these bones. There are fairly standard methods to measure mechanical strength. Having such data would make the case much stronger that blockade of PHD2 function increases functional bone density.

To investigate whether the increase in trabecular and cortical bone mass, determined by histological and microCT analysis, results in altered bone mechanical strength, we performed three-point-bending tests on femurs of 8-week-old mice. We detected a significant increase in ultimate force, work-to-failure and stiffness (revised Supplementary Figure 2c-e), indicating that *Phd2*^{ot-} bones are stronger than wild type. Importantly, post-yield displacement was similar between genotypes (revised Supplementary Figure 2f), suggesting that osteocytic deletion of PHD2 does not affect matrix composition or organization (Jepsen, KJ JBMR 2015). Together, these results, discussed in the revised manuscript on page 6, show that the enhanced bone mass in mutant mice is associated with increased biomechanical strength.

2- I appreciate the challenge of finding something that simulates osteoporosis by restricting leg use or by changing estrogen status. However, such measures are relatively acute and may not reflect the kinetics of bone loss in a clinically relevant manner. Along the same lines, the studies reported in this paper were with very young mice (3 week). Having some studies performed in aged mice would strengthen the arguments made. Any results seen would be closer to the most common clinical scenario of bone loss after menopause.

We fully agree with the reviewer's comment that bone loss observed in aged animals mimics more closely the clinical scenario. Therefore, we compared trabecular bone mass (BV/TV) and cortical thickness (Ct.Th) in *Phd2*^{ot+} and *Phd2*^{ot-} mice at several time points using microCT. During bone growth, both male and female *Phd2*^{ot-} mice showed a trend toward increased bone mass,

starting already at three weeks of age. This high bone mass phenotype became more apparent in adult mice and persisted with ageing. Importantly, whereas bone mass in *Phd2^{0t+}* mice progressively declined with age, BV/TV and Ct.Th in *Phd2^{0t-}* mice remained high, indicating that osteocytic deletion of PHD2 protects mice from age-related bone loss.

In addition, we regret to have caused confusion regarding the age of the mice that were used in the different experiments. For clarification: basic bone phenotyping (revised Figures 1, 2, 7 and Supplementary Figures 1, 3, 4) was performed in 8-week-old male mice; the effects on bone by modulation of SIRT1 signalling (revised Figure 5 and Supplementary Figure 10) were investigated in 8-week-old male mice; and the analysis of the bone phenotype after hindlimb unloading (revised Figure 8 and Supplementary Figure 11) or ovariectomy (revised Figure 9 and Supplementary Figure 12) was performed in 20-week-old or 14-week-old mice, respectively. We have included this information in the Methods section and in the Figure legends.

Specific Comments

1- Many western blots are shown in the figures. As far as I can tell, there did not appear to be any quantification of results. This would be quite useful, as it would give the reader some idea of the magnitude of changes in proteins being assessed.

As suggested, we have now quantified all immunoblots. Please note that because of incorporation of additional data, figure panels are changed: Figure 4j is now Figure 5j, Figures 5a-b are Figures 6a-b, Figure 7j is Figure 8j, Figure 8j is Figure 9j.

2- There is increasing emphasis on using both male and female mice in biomedical research. I was unable to tell whether both male and female mice were used and importantly, whether there were differences observed. I realize that estrogen loss is not really relevant to male mice, but lack of weight bearing would be relevant.

We thank the reviewer for pointing this out. We have added this information to the Methods section (p. 20-21).

REVIEWERS' COMMENTS:

Reviewer #1 (Remarks to the Author):

Overall, the authors have satisfactorily addressed my comments. They have to be commended for an excellent revision with abundant new data.

Reviewer #2 (Remarks to the Author):

I feel authors adequately addressed my suggestions and concerns.

Reviewer #3 (Remarks to the Author):

I was quite enthusiastic about this paper in the first review. However, I had some high level concerns:

1- The assessment of PHD2 on bone formation had focused primarily on histologic analysis. I encouraged the authors to provide a secondary analysis, testing the mechanical strength of the bones. The authors responded positively to this suggestion and performed the suggested studies. The results were consistent with the histologic data.

2- I was under the impression that most of the studies had been performed in young mice, which does not reflect the underlying biology that might be associated with aging. The authors also responded very positively to this issue, by performing bone mass studies across the age continuum, using microCT. Here they showed that Phd2^{-/-} mice showed a trend toward increased bone mass compared with wild type mice, across the age continuum. By comparison, there was a significant decline in bone mass in wild type mice with aging.

The authors further clarified the range of mouse ages that were used in this study.

3- I suggested that the authors perform quantitative analysis of western blots. This was done.

4 - Finally, I asked for clarification as to whether both male and female mice were used. This point was also clarified.

Overall, I believe that the authors have provided very convincing evidence that PDH2 is involved in regulation of bone mass by regulation of Sirtuin and sclerostin. This work sheds important new light on this very important hormone driven consequence of aging. As such, it provides important new clues for pharmacologic targeting of osteoporosis.

We thank the reviewers for their positive feedback, stating that we addressed their comments and suggestions adequately. No additional concerns or clarifications were raised.

Reviewer #1 (Remarks to the Author):

Overall, the authors have satisfactorily addressed my comments. They have to be commended for an excellent revision with abundant new data.

Reviewer #2 (Remarks to the Author):

I feel authors adequately addressed my suggestions and concerns.

Reviewer #3 (Remarks to the Author):

I was quite enthusiastic about this paper in the first review. However, I had some high level concerns:
1- The assessment of PHD2 on bone formation had focused primarily on histologic analysis. I encouraged the authors to provide a secondary analysis, testing the mechanical strength of the bones. The authors responded positively to this suggestion and performed the suggested studies. The results were consistent with the histologic data.

2- I was under the impression that most of the studies had been performed in young mice, which does not reflect the underlying biology that might be associated with aging. The authors also responded very positively to this issue, by performing bone mass studies across the age continuum, using microCT. Here they showed that Phd2^{-/-} mice showed a trend toward increased bone mass compared with wild type mice, across the age continuum. By comparison, there was a significant decline in bone mass in wild type mice with aging.

The authors further clarified the range of mouse ages that were used in this study.

3- I suggested that the authors perform quantitative analysis of western blots. This was done.

4 - Finally, I asked for clarification as to whether both male and female mice were used. This point was also clarified.

Overall, I believe that the authors have provided very convincing evidence that PDH2 is involved in regulation of bone mass by regulation of Sirtuin and sclerostin. This work sheds important new light on this very important hormone driven consequence of aging. As such, it provides important new clues for pharmacologic targeting of osteoporosis.